# High throughput, label-free isolation of circulating tumor cell clusters in meshed microwells

Mert Boya[1], Tevhide Ozkaya-Ahmadov[1], Brandi E. Swain[1], Chia-Heng Chu[1], Norh Asmare[1], Ozgun Civelekoglu [1], Ruxiu Liu [1], Dohwan Lee [1], Sherry Tobia[2], Shweta Biliya[3], L. DeEtte McDonald[3,4], Bassel Nazha[5,6], Omer Kucuk[5,6], Martin G. Sanda[7], Benedict B. Benigno[8,9], Carlos S. Moreno [5,10], Mehmet A. Bilen[5,6], John F. McDonald[3,4,8,9] & A. Fatih Sarioglu [1,3,5,8,11✉]

Extremely rare circulating tumor cell (CTC) clusters are both increasingly appreciated as highly metastatic precursors and virtually unexplored. Technologies are primarily designed to detect single CTCs and often fail to account for the fragility of clusters or to leverage cluster-specific markers for higher sensitivity. Meanwhile, the few technologies targeting CTC clusters lack scalability. Here, we introduce the Cluster-Wells, which combines the speed and practicality of membrane filtration with the sensitive and deterministic screening afforded by microfluidic chips. The >100,000 microwells in the Cluster-Wells physically arrest CTC clusters in unprocessed whole blood, gently isolating virtually all clusters at a throughput of >25 mL/h, and allow viable clusters to be retrieved from the device. Using the Cluster-Wells, we isolated CTC clusters ranging from 2 to 100+ cells from prostate and ovarian cancer patients and analyzed a subset using RNA sequencing. Routine isolation of CTC clusters will democratize research on their utility in managing cancer.

[1] School of Electrical and Computer Engineering, Georgia Institute of Technology, Atlanta, GA, USA. [2] University Gynecologic Oncology, Atlanta, GA, USA. [3] Parker H. Petit Institute for Bioengineering and Bioscience, Georgia Institute of Technology, Atlanta, GA, USA. [4] School of Biological Sciences, Georgia Institute of Technology, Atlanta, GA, USA. [5] Winship Cancer Institute, Emory University, Atlanta, GA, USA. [6] Department of Hematology and Medical Oncology, Emory University School of Medicine, Atlanta, GA, USA. [7] Department of Urology, Emory University School of Medicine, Atlanta, GA, USA. [8] Integrated Cancer Research Center, Georgia Institute of Technology, Atlanta, GA, USA. [9] Ovarian Cancer Institute, Atlanta, GA, USA. [10] Department of Pathology and Laboratory Medicine, Emory University School of Medicine, Atlanta, GA, USA. [11] Institute for Electronics and Nanotechnology, Georgia Institute of Technology, Atlanta, GA, USA. ✉email: sarioglu@gatech.edu

Single circulating tumor cells (CTCs) harvested from the bloodstream of cancer patients provide valuable information on the stage of the disease[1], enable minimally invasive prognosis and diagnosis[2–4], enhance our understanding of metastasis at the cellular level[5,6], and offer the potential to improve the clinical management of cancer[7,8]. In addition to these single CTCs, CTC aggregates that remain attached in circulation have been of great scientific and clinical interest since the 1950s. Although these CTC clusters, or circulating tumor microemboli, are extremely rare (estimated at only 2–5% of all CTCs), they are disproportionately efficient at seeding metastases. Their metastatic propensity is estimated to be 100 times higher than that of single CTCs[9–11], based on their lower apoptosis rate and reduced prolonged survival attributes[9,12]. Moreover, a subset of patient-derived CTC clusters was found to include host immune cells[13,14], highlighting the utility of these metastatic precursors to shed light on tumor–immune system interactions and their role in metastasis. For example, CTC clusters carrying neutrophils have been shown to have increased metastatic potential in advanced breast cancer patients[14], where the neutrophil-associated CTCs demonstrate higher expression levels of proliferation marker protein (Ki67) and of genes associated with cell-cycle progression. Clinical studies have supported the results of these biological investigations, finding that the presence of CTC clusters is associated with shorter progression-free survival and overall patient survival[15]. The increased study of CTC clusters, then, offers great promise in improving the understanding and management of metastasis.

To date, reliable and efficient isolation of viable CTC clusters has been limited because the sensitivity and specificity of CTC isolation technologies are primarily calibrated for single cell detection[16–19]. Microfiltration techniques, for example, are widely employed as CTC assays due to their fast and straightforward operation[16,19,20]. However, CTC clusters under physiological pressure can reorganize themselves as single-file chain-like structures and traverse constrictions as small as 5 μm[11], which suggests that aggregates can likely pass through filter pores, given the much higher pressures employed in filtration. Moreover, higher shear forces experienced during microfiltration could damage CTC clusters or dissociate them into single cells[21], undermining efficient enrichment. On the other hand, antibody-based enrichment systems, which have long been used for isolation of single CTCs[17,22,23], can only detect a select sub-population of CTC clusters within the heterogenous CTC population due to their dependency on specific membrane antigens[13,24]. Additionally, the smaller surface-area-to-volume ratio of CTC clusters negatively impacts the immunocapture efficiencies of antibody-based technologies[25], rendering them particularly inefficient for CTC cluster enrichment. More promising in this regard are recent microfluidic chips specifically targeting CTC clusters, which can achieve relatively higher sensitivities. Unfortunately, they do so either at clinically unworkable processing rates[21,26] or under threat of cluster damage due to the high flow rates in narrow channels[27],—a concern especially relevant for large clusters, which were reported to be occasionally comprised of as many as tens of tumor cells[28].

To address these limitations for the isolation and study of CTC clusters, we have developed the Cluster-Wells. This high-sensitivity, high-throughput polymer device gently isolates CTC clusters, small and large, from unprocessed whole blood specimens without the need for targeting tumor-specific antigens. Instead, the Cluster-Wells retains CTC clusters in microwells engineered to physically recognize cellular aggregates and preserve their integrity under sub-physiological flow rates. Moreover, unrestricted access to purified CTC clusters in microwells permits further investigation via imaging, and functional and molecular assays.

## Results

**Design and microfabrication of the Cluster-Wells.** The Cluster-Wells captures CTC clusters in microwells designed to retain cell aggregates while being permeable to single cells in blood (Fig. 1a). At the bottom of each well is a micromesh with openings of

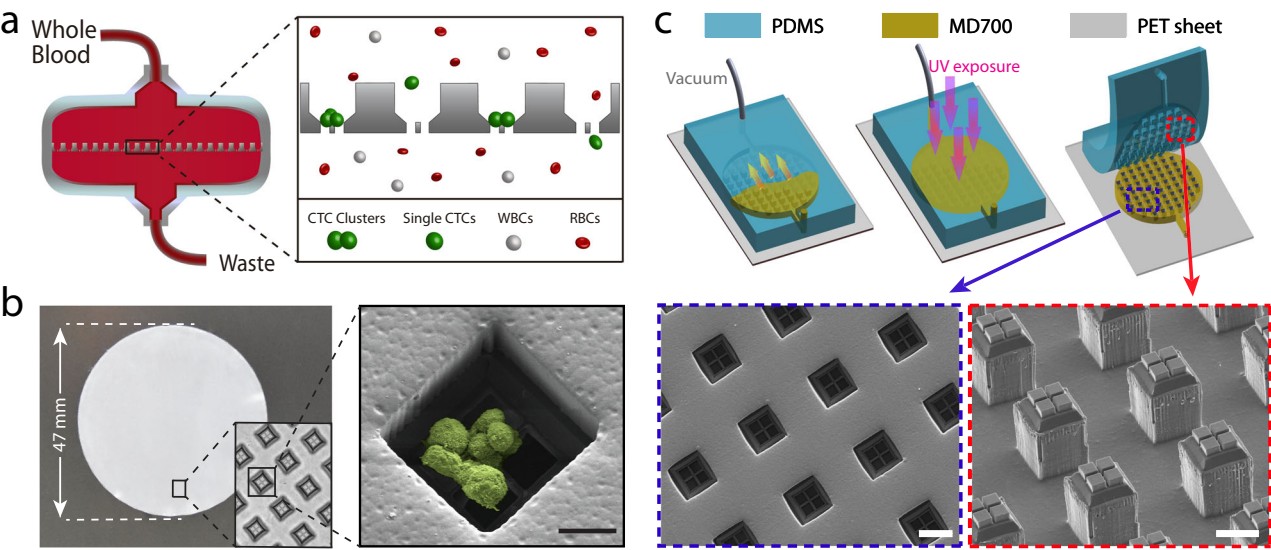

**Fig. 1 Design and microfabrication of the Cluster-Wells. a** Schematic illustration of the Cluster-Wells' working principle. While single cells pass unimpeded, the Cluster-Wells captures CTC clusters owing to their multicellular morphology from blood samples of cancer patients, independent of cancer type and their molecular character. **b** A photo of the Cluster-Wells manufactured in the form of a 47 mm-diameter membrane to be used in commercial filter holder. Close-up image (inset) shows individual wells designed to capture CTC clusters. (Right) Scanning electron micrograph of a blood-spiked LNCaP cluster as captured by one of the wells on the device (see "Methods": SEM sample preparation and imaging). Scale bar, 20 μm. **c** Schematic illustration of the fabrication process developed to mold polymer Cluster-Wells device. Steps shown in the figure exclude the fabrication process for the silicon mold and the construction of reusable PDMS mold, which can be found in Supplementary Figs. 2 and 3. Scanning electron micrographs of the PDMS structure employed for micromolding (lower right) and the finished device fabricated out of photocurable polymer, MD700 (lower left). Scale bars, 50 μm.

15 μm × 15 μm, a size that lets leukocytes, erythrocytes, and platelets pass unimpeded. Thus, unprocessed whole blood flows easily through the device, but CTC clusters, too big to commit to any single opening, are captured when individual cells within the cluster lodge in neighboring mesh openings. Once inside a well, CTC clusters are constrained on the mesh by slanted sidewalls, which restrict potential reorientation under flow and protect the cells from the damaging transverse stresses found in traditional membrane filters.

The mesh lines are designed to be thin (~2 μm-wide), so they function as a wedge between cells, forcing cluster cells into different openings and arresting the cluster at cell–cell junctions. To guard against cluster dissociation by mesh lines, cell flow speed was set as low as 65 μm/s (Supplementary Fig. 1), ~10× lower than physiological free flow speed in human capillaries[29]. To maintain gentle handling of CTC clusters while still achieving clinically relevant throughput rates, we operated a large number of cluster-trapping wells in parallel. Even at the specified flow rate, 120,000 microwells uniformly spread over a 47 mm diameter membrane provided a processing rate of 25 mL/h (Fig. 1b).

We designed the Cluster-Wells as a disposable polymer device to ensure quick, low-cost manufacturing that would make the technology accessible in a variety of research and clinical settings. The microfabrication process combined silicon micromachining[30], soft lithography[31], and micromolding techniques[32] (see "Methods": microfabrication of the silicon master-mold & molding of the Cluster-Wells from the silicon master). We first manufactured a negative mold out of silicon using a 3-mask microfabrication process that involved thin film deposition with wet and reactive ion etching steps (Supplementary Fig. 2). The silicon mold was then transferred into polydimethylsiloxane (PDMS) and replicated using soft lithography. The resulting negative PDMS mold was filled with a photocurable polymer (Fluorolink MD700, Solvay) and crosslinked in the mold to form the device (Fig. 1c). The fabricated device was 65 μm thick and structurally rigid, allowing for easy handling. This process allows us to transfer the intricate geometry only achievable with silicon micromachining to affordable and disposable plastic devices.

**Device characterization and optimization using simulated blood samples**. To test and optimize the operation of Cluster-Wells, we processed samples prepared by spiking artificially formed clusters of human cancer cell lines into blood samples collected from healthy donors according to an IRB-approved protocol (see "Methods": cell culture and preparation & sample collection). In these experiments, tumor cell clusters were either pre- or post-labeled with fluorescent dyes for positive identification. Cluster-spiked whole blood was driven through a Cluster-Wells placed in a commercial filter holder, followed by a wash with phosphate-buffered saline (PBS) and immunofluorescent staining of cells on the device. The captured clusters were identified in microwells via fluorescence microscopy (Fig. 2a).

To determine the capture efficiency of the Cluster-Wells under different flow rates, we imaged and compared the spiked cluster populations entering and exiting the device. We built a 2-channel microfluidic interface to image each processed cluster under a microscope, which allowed us to simultaneously track fluorescent cancer cells in whole blood during their entry and exit within the same field of view (Fig. 2b) (see "Methods": measurement of the device sensitivity). For these experiments, we stained cell nuclei rather than cell membranes or cytoplasm to provide better visual distinction between cells and facilitate more accurate enumeration of cells within clusters. Additionally, the imaging speed was set high enough to (1) secure multiple (>4) shots of each cluster, ensuring none of the clusters were missed and (2) capture multiple different

conformations within the field of view, increasing the accuracy of cluster size estimation. We validated (see "Methods": measurement of the device sensitivity) the optimized characterization setup by operating the 2-channel microfluidic interface in a loop without the device attached to confirm there were no cell loss in the system (Supplementary Table 1) and also by ensuring the cluster counts obtained using our setup matched with the direct counts of the captured clusters on the device (Supplementary Fig. 4).

Using the Cluster-Wells, we processed whole blood samples spiked with clusters of human prostate cancer cells (LNCaP) at rates ranging from 25 to 750 mL/h (Fig. 2c). At 25 mL/h, the device isolated nearly all clusters, missing only ~6.8% of doublets. At 250 mL/h, a throughput that allows screening of a tube of whole blood in only ~2 mins, the cells still flowed through the microwells at physiological flow speeds, and the device captured all of tumor cell clusters consisting of 6 or more cells, ~98.7% of 5-cell clusters, ~93.8% of 4-cell clusters, ~89.3% of 3-cell clusters, and ~75.8% of doublets. Only at flow rates >500 mL/h, spiked cell clusters dissociated in the device, as evidenced by the mismatch observed between the size distributions of spiked and processed cell populations (Supplementary Fig. 5).

To investigate the device's sensitivity when processing tumor cell clusters from different cancer types, we also processed samples spiked with clusters of breast (MDA-MB-231 and MCF-7) and ovarian (HeyA8) cancer cell lines. These tumor cells have different biochemical and biophysical properties than the previously tested prostate cancer cells, as well as differences in cell-to-cell affinities. Even so, at 25 mL/h the Cluster-Wells captured 90% or more of doublets, the most evasive cluster type, for all tumor cell lines, which supports the applicability of this technology to cancers of different tissues (Fig. 2d). To put the measured sensitivity of the device into perspective: the Cluster-Wells captures doublets with twice the reported sensitivity of a similarly-sized Cluster-Chip[21], a device that was already shown to be more sensitive for capturing clusters than techniques optimized for single CTCs, while screening the blood 10× faster (Supplementary Fig. 6). In fact, when both platforms screen samples at the same flow rate (50 mL/h), the Cluster-Wells captures the smallest clusters with efficiency almost an order of magnitude (~9×) higher based on the published capture rates[21] of the Cluster-Chip. Even when screening whole blood at a throughput of 100 mL/h, the Cluster-Wells is more sensitive than every CTC cluster enrichment method, only missing ~10.4% of doublets and ~2.7% of triplets.

In order to optimize the device geometry for CTC cluster isolation, we investigated the effect of micromesh opening size on the device performance. In addition to the Cluster-Wells described in this paper, we designed a version with smaller (13.5 μm) openings, and another with larger (16.5 μm) openings. Testing again with whole blood spiked with human prostate cancer cells (LNCaP), we found that the adjusted opening sizes presented trade-offs between cluster capture sensitivity and specificity. While each device, tested at 25 mL/h, could capture all of the larger (4+ cells) clusters, 16.5 μm-mesh resulted in a ~20% decrease in doublet capture efficiency and a ~2.5% decrease in 3-cell cluster capture (Fig. 2e). Conversely, decreasing the opening to 13.5 μm increased the doublet capture efficiency ~5%, for a total of ~98% capture. However, the smaller mesh opening also decreased specificity, allowing ~60% higher leukocyte contamination on average (Fig. 2f). However, since most leukocytes are not mechanically trapped but rather adhere to surfaces non-specifically at random locations, increasing the mesh openings offered diminishing gains in specificity: larger (16.5 μm) mesh openings decreased leukocyte contamination by only ~20%. Weighing both the purity and capture efficiency data gathered, we have identified the 15 μm mesh opening as the

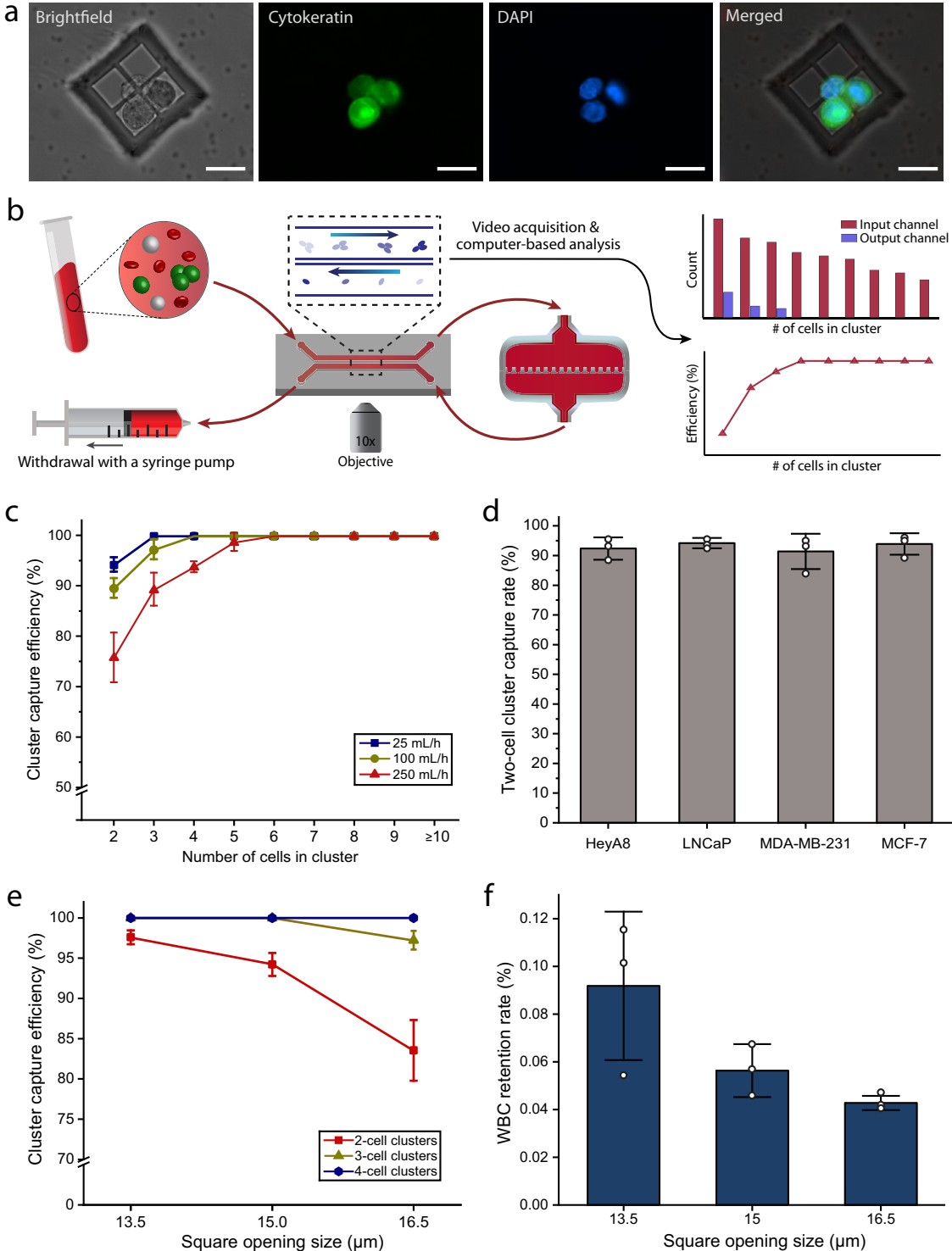

optimum design choice, which yielded an average contamination of ~28.8 white blood cells (WBC)/mm$^2$ (<0.06%) along with ~331 platelets/mm$^2$ (<0.025%) from the processing of a 10 mL of unmanipulated whole blood sample in our tests.

**Cluster retrieval from the Cluster-Wells**. Because the release of viable CTC clusters is crucial for downstream molecular and functional assays, we built the Cluster-Wells out of a fluorine-based low surface energy polymer (Fluorolink MD700, Solvay) to improve cluster retrieval by minimizing nonspecific cell adhesion, a documented drawback to PDMS-based devices[21]. To assess the

Cluster-Wells' release performance, we attempted to retrieve viable tumor cell clusters captured on the device (Fig. 3a). After processing whole blood samples spiked with LNCaP clusters at 25 mL/h, we washed the Cluster-Wells with PBS and released the clusters into Petri dishes at varying reverse flow speeds. Using fluorescence microscopy to identify both the clusters successfully released in the collecting Petri dish and the ones that remained on the device under each tested speed, we calculated cluster release efficiencies to determine the optimum reverse flow speed. The release efficiency increased with higher reverse flow speeds, which displaced more of the clusters non-specifically adhered to the

**Fig. 2 Characterization of the Cluster-Wells using unprocessed whole blood. a** Representative fluorescence images of a captured LNCaP cluster spiked into unprocessed whole blood. Following the PBS wash and fixation, the isolated tumor cells were stained with Cytokeratin (green) and DAPI (nuclei, blue). Scale bars, 20 μm. **b** Schematic illustration of the experimental setup used for the characterization studies. The cells entering and exiting the analytical versions of the Cluster-Wells were visually tracked and counted for capture efficiency calculations using a 2-channel microfluidic interface (see "Methods": measurement of the device sensitivity). **c** Cluster capture efficiencies for the spiked LNCaP clusters at different flow rates. Data is provided according to the number of cells in clusters ($n = 3$ independent experiments). Data are presented as mean ± SD. **d** Two-cell cluster capture rates for prostate (LNCaP), breast (MDA-MB-231 & MCF-7), and ovarian (HeyA8) cancer cell clusters at the flow rate of 25 mL/h ($n = 3$ independent experiments). Data are presented as mean ± SD. Since the Cluster-Wells solely relies on physical attributes of clusters, it efficiently captured clusters of different tumor types, independent of their surface antigens. **e** Two, three, and four-cell LNCaP cluster capture efficiencies for the devices with different opening sizes ($n = 3$ independent experiments). Experiments were performed at 25 mL/h flow rate. Data are presented as mean ± SD. **f** White blood cell (WBC) retention rates for the devices with different opening sizes after processing a tube (10 mL) of unprocessed whole blood ($n = 3$ independent experiments). The non-specifically attached WBCs were stained and counted. For the calculation of retention rate, obtained WBC count was divided by the total number of WBCs ran through the device, which was acquired from a complete blood count (CBC) analyzer. Data are presented as mean ± SD. Source data are provided as a Source Data file.

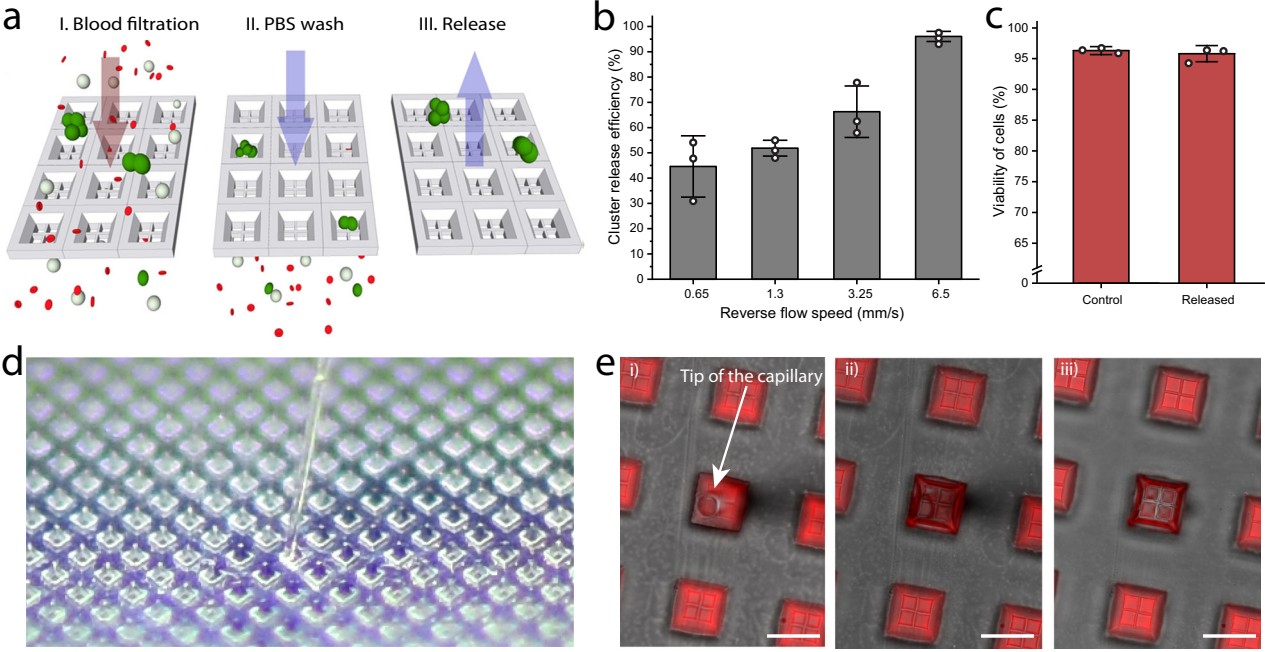

**Fig. 3 Retrieval of captured clusters from the Cluster-Wells. a** Representation of the release process. LNCaP clusters were spiked into whole blood and processed using Cluster-Wells. Following the PBS wash, captured clusters were released into a Petri dish by the application of reverse flow. **b** Release efficiency of the device at different relative reverse flow speeds. Following the release, both Petri dish and device were imaged using a fluorescence microscope, and the cells were counted manually for the calculation of release efficiency ($n = 3$ independent experiments). Data are presented as mean ± SD. **c** Percentage viability of control and clusters released into a Petri dish at 6.5 mm/s reverse flow speed ($n = 3$ independent experiments). The two-color viability test was performed on the control and released cells inside the Petri dishes (see "Methods": measurement of the cell viability), which were subsequently scanned using fluorescence microscope. The number of viable and dead cells were counted for calculating the percentage of viable cells. Data are presented as mean ± SD. **d** A photo of the Cluster-Wells and the capillary of a micromanipulator inside an individual microwell. Unlike microfluidic devices, individual microwells could easily be accessed using a micromanipulator. **e** Fluorescence microscope images of the Cluster-Wells after being removed from the filter holder. Due to surface tension, previously introduced fluorescent dye solution stays inside the wells and the content of individual microwells can be retrieved using a micromanipulator. When left unattended, microwells dry in ~ 10 min. Scale bars, 50 μm. Source data are provided as a Source Data file.

device. We concluded that under an average reverse flow rate of ~6.5 mm/s within the microwells (100× the capture flow speed) for ~10 seconds, almost all (~96%) of the clusters could be retrieved successfully (Fig. 3b). Furthermore, we confirmed the integrity of released clusters was preserved by directly comparing the sizes of captured and released cluster populations (Supplementary Fig. 7).

The measured retrieval efficiency of the Cluster-Wells is higher than the previously reported cluster retrieval rate of PDMS-based Cluster-Chip[21]. Mimicking the testing conditions reported for the Cluster-Chip in our experiments, we used the Cluster-Wells to process blood samples spiked with MDA-MB-231 clusters and

attempted to release the captured aggregates. Using the same reverse flow speed (6.5 mm/s) and duration previously applied with the Cluster-Chip, we were able to retrieve clusters from Cluster-Wells with >2× (87% vs 37%) the efficiency of the Cluster-Chip (Supplementary Fig. 8). Even when compared to the reported release efficiency of a Cluster-Chip operated at 4 °C to reduce non-specific cell adhesion to the microfluidic chip, the Cluster-Wells demonstrated greater efficiency (87% retrieval for the Cluster Wells vs 80% for the Cluster-Chip).

Next, using a live/dead cell assay, we investigated the viability of tumor cells captured and subsequently released from the Cluster-Wells (see "Methods": measurement of the cell viability)

(Supplementary Fig. 9). For these measurements, LNCaP clusters captured from spiked blood samples by the Cluster-Wells were released under reverse flow at 6.5 mm/s. The average viability of released clusters (~95.8%) was found to be similar to that of the control population (~96.3%), demonstrating neither the device nor the release protocol produced a prominent effect on cell viability (Fig. 3c).

To evaluate whether we could take advantage of the exposed layout of the Cluster-Wells to retrieve cells, we also investigated whether viable clusters could be captured directly from microwells using a micromanipulator (Fig. 3d). First, we tested whether captured clusters could be safely retained in the microwells while the device was set up for micromanipulation. By comparing the number of clusters imaged on the exposed device with the number of spiked clusters, we found that the majority (~95.8%) of clusters were retained on the exposed device. We then ran a separate test to determine whether the device with its porous microwells would dry prematurely once taken out of the filter holder and exposed, which would prohibit the collection of viable clusters. We filled the Cluster-Wells with a fluorescent dye solution and found that the hydrophobicity of the device combined with surface tension prevented liquid from leaking through the micromesh under gravity (Fig. 3e). Moreover, as the liquid evaporated over the surface of the exposed device, the microwells were last to dry, providing a natural isolation between wells so that their contents could be collected without crosstalk (Fig. 3e).

**Isolation of CTC clusters from patient samples**. To assess the device's potential clinical application, we applied the Cluster-Wells technology to isolate CTC clusters from blood samples of ovarian and prostate cancer patients. The blood samples, ranging in volume from 2.8 to 24 mL, were collected from consenting patients according to IRB-approved protocols at Emory University and Northside Hospitals and processed at Georgia Tech within 4 h of collection (see "Methods": sample collection & sample processing). Following the fixation of isolated cells with 4% paraformaldehyde (PFA), samples were immunostained with leukocyte markers and against established cancer markers for the specific disease; a nuclear stain (4,6-diamidino-2-phenylindole) was also applied to positively identify patient CTC clusters (see "Methods": immunofluorescence staining of CTC clusters). Immunostained clusters were then imaged and analyzed by scanning the Cluster-Wells under a fluorescence microscope. The specificity of the assay was also validated for both cancer types by processing 10 mL of healthy donor blood samples ($n = 3$ each), which were all confirmed to be absent of CTC clusters.

We detected CTC clusters in both ovarian (Fig. 4a) and prostate (Fig. 4b) cancer patient blood samples. The number of cells in these clusters ranged from 2 to as high as >150 cells, with the latter found in a sample from an ovarian cancer patient (Fig. 4a, iii). Besides raising questions on the physiological circulation of CTC clusters, the surprisingly large size of the isolated ovarian cancer CTC cluster points to a potential drawback of microfluidic chips, even if they are designed with CTC clusters in mind: a CTC cluster of this size would have likely clogged narrow microfluidic channels or split into smaller pieces if forced through. Some of the isolated CTC clusters were also found to contain leukocytes in both ovarian and prostate cancer samples (Fig. 4a, ii and b, iii), an observation consistent with previous reports on breast cancer CTC clusters[14]. To confirm physical cohesion between cells in these leukocyte-associated CTC clusters, such clusters were purposely expelled from microwells and then recaptured.

To test the applicability of the Cluster-Wells in isolating cellular aggregates from bodily fluids other than whole blood, we attempted to isolate tumor spheroids from an ascites sample collected from an ovarian cancer patient (see "Methods": sample processing). The concentration of tumor cell spheroids observed in ascites sample was significantly higher compared to that of CTC clusters observed in a peripheral blood sample collected from the same patient at the same timepoint (~1500 spheroids/mL vs 0.167 CTC clusters/mL). Moreover, tumor cells in isolated spheroids were observed to be morphologically larger than those in CTC clusters that we isolated from the blood sample (Fig. 4c and Supplementary Fig. 10).

Among the patient cohort we studied (Supplementary Tables 2 and 3), CTC clusters were detected in 6/8 patients with prostate cancer and in 8/9 patients with ovarian cancer (Fig. 4d). The number of detected CTC clusters varied widely between patients, with concentrations ranging from 0.063 to 0.867 clusters/mL in prostate cancer patients and from 0.167 to 59.2 clusters/mL in ovarian cancer patients (Fig. 4e). On average, ovarian cancer CTC clusters were composed of more tumor cells than prostate cancer CTC clusters with a median of 6 cells versus 3 cells in prostate cancer (Fig. 4f). Moreover, 26.4% of the isolated ovarian cancer CTC clusters contained WBCs, while 16.4% of prostate cancer CTC clusters were found to contain WBCs (Fig. 4g). Taken together, the observed phenotypical heterogeneity in CTC clusters isolated by the Cluster-Wells attested to the high dynamic range and gentle processing conditions offered by the technique. Also, a higher concentration (0.43/mL vs 0.28/mL[21]) of prostate cancer clusters detected by the Cluster-Wells compared to the Cluster-Chip, albeit in different cohorts, could potentially be due to the superior isolation efficiency of the Cluster-Wells, given the small size of prostate cancer CTC clusters.

Importantly, CTC clusters were notably more prevalent in the ovarian cancer patient cohort, with an average concentration of 10.32 clusters/mL. The high occurrence of CTC clusters in ovarian cancer patients, particularly in light of the previous studies on single CTCs in ovarian cancer[33], suggests highly efficient intravasation capabilities of ovarian cancer cells. In addition, by investigating the viability of CTC clusters from one of the ovarian cancer patients (see "Methods": viability assessment of CTC clusters isolated from patient samples), we found that the CTC clusters isolated by Cluster-Wells contained viable tumor cells (Fig. 4h), which further highlighted the advantages of the rapid, yet gentle enrichment provided by the technology. Overall, the isolation of viable CTC clusters, therefore, might be particularly useful for the diagnosis and surveillance of ovarian cancer[34].

**Molecular analysis of patient CTC clusters**. To demonstrate the feasibility of using Cluster-Wells to isolate viable CTC clusters for downstream molecular assays, we performed RNA sequencing on individual CTC clusters from two prostate cancer patients (see "Methods": RNA library preparation and sequencing & RNA sequencing data analysis). Unfixed cells on the Cluster-Wells were first immunostained against tumor-specific membrane antigens and leukocyte markers, and fluorescent CTC clusters were picked directly from the microwells for sequencing using a micromanipulator (Fig. 5a) (see "Methods": micromanipulation of CTC clusters from patient samples). As the samples for molecular studies were subjected to a different staining protocol that exclusively targeted surface markers (Fig. 5b), they were not included in enumeration of CTC clusters in prostate cancer patients.

A total of 6 CTC clusters were sequenced from two prostate cancer patients, as well as two sets of WBC, and two clusters from LNCaP and PC-3 prostate cancer cell lines each. From the patient clusters, we obtained a median of 83.15 M reads per sample, with a median of 74 M uniquely mapped reads (~88.8%), and an

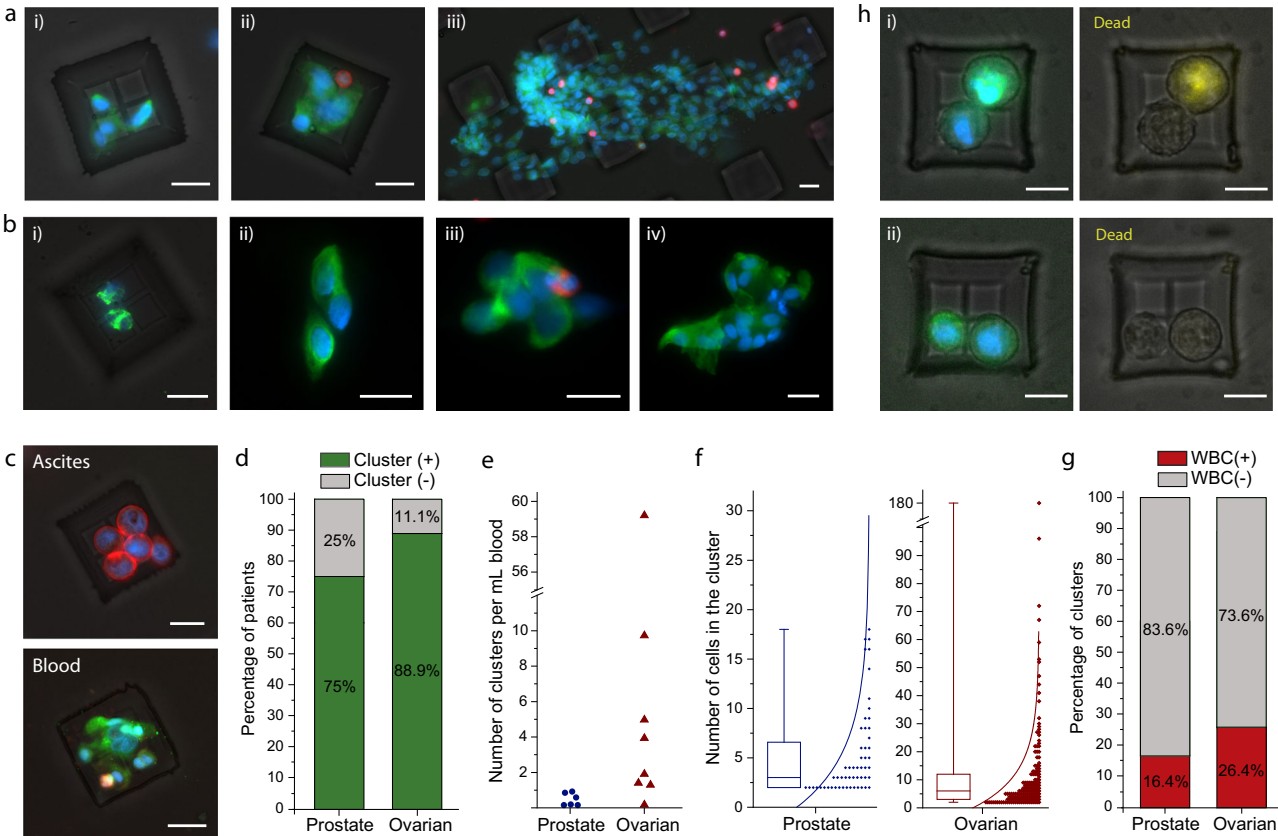

**Fig. 4 Isolation of CTC clusters from ovarian and prostate cancer patients. a** Fluorescence microscope images of the isolated CTC clusters from ovarian cancer patients. Captured cells were stained with Cytokeratin, Vimentin (green), CD45 (red), and DAPI (nuclei, blue). Scale bars, 20 μm. **b** Images of the isolated CTC clusters from metastatic prostate cancer patients. Captured cells were stained with Cytokeratin, Vimentin, PSA/KLK3, EpCAM (green), CD45(red), and DAPI (nuclei, blue). Scale bars, 20 μm. **c** Clusters isolated from the peripheral blood and ascites sample drawn from the peritoneal cavity of an ovarian cancer patient at the same timepoint. The isolated CTC cluster was morphologically different compared to the tumor spheroids isolated from ascites sample of the same patient. The spheroids from ascites sample were fixed and stained with Cytokeratin, EpCAM (red), CD45 (green) and DAPI (nuclei, blue), and CTC clusters isolated from blood sample were fixed and stained with Cytokeratin, Vimentin (green), CD45 (red) and DAPI (nuclei, blue). Scale bars, 20 μm. **d** Percentage of prostate (*n* = 8) and ovarian (*n* = 9) cancer patients that are observed to have CTC cluster(s) in their blood samples. **e** Number of CTC clusters observed per milliliter of blood samples from prostate and ovarian cancer patients. **f** Distribution of the number of cells observed in the prostate and ovarian CTC clusters. The box plots show the 25th and 75th percentiles, line shows median, and whiskers show maxima and minima points. **g** Percentage of pure and white blood cell associated CTC clusters for each disease type. **h** Images of the CTC clusters isolated from an ovarian cancer patient and subjected to a fluorescence-based viability assay. The images show CTC clusters that were composed of (i) both live and dead (yellow) cells, and (ii) live cells only. Scale bars, 20 μm. Source data are provided as a Source Data file.

average Phred score of 36 for mean sequence quality. We detected a total of 16,412 unique transcripts and used the entire set of detected transcripts to perform a t-SNE analysis to compare the patient clusters to the control WBCs and prostate cancer cell lines (Fig. 5c) and observed that different clusters from the same patient or cell line tended to cluster together.

First, we performed DESeq2 analysis[35] comparing the patient clusters to the WBCs to identify differentially expressed transcripts and identified a total of 3718 transcripts (p-adj < 0.01). The entire ranked list of detected transcripts was used for gene set enrichment analysis[36] using WebGestalt[37], and we observed numerous highly significant gene sets enriched in the patient clusters including MYC target genes, G2M checkpoint genes, E2F target, and Estrogen Response genes (Supplementary Figs. 12 and 13) from the Hallmark 50 gene sets and the KEGG pathway gene set collections. These gene sets are consistent with proliferative, hormonally driven cells such as malignant tumor cells[38–40]. In addition, the patient clusters were negatively enriched for immune cell-related gene sets such as natural killer cell cytotoxicity genes, graft vs host disease genes, interferon gamma response genes, and Th1/Th2 differentiation genes, among others, suggesting that the transcriptome of these

clusters differed from healthy immune cells of the host (Supplementary Fig. 14).

Analysis of transcripts corresponding to a selected set of genes for prostate cancer showed all CTC clusters expressed high levels of epithelial markers (Fig. 5d). Among the epithelial markers, Epithelial Cell Adhesion Molecule (*EpCAM*), E-Cadherin (*CDH1*), and Cytokeratin (*KRT*) *8, 18*, and *19* were highly expressed in all of the CTC clusters, while notably CTC clusters from Patient-2 also expressed *KRT5*. Among the mesenchymal markers, Vimentin (*VIM*), an indication of induction of epithelial-to-mesenchymal transition (EMT)[41], was expressed by most of the CTC clusters at relatively lower but detectable levels. In addition to established tumor markers, patient clusters were also screened for stem cell and proliferation-associated markers. Among the stem cell markers, *CD24* were expressed in all of the CTC clusters at varying levels, while 5/6 clusters were found to express high levels (>50 rpm) of *CD44*. It has been previously reported that *CD44* homophilic interactions and subsequent *CD44–PAK2* interactions mediate tumor cell aggregation[42] and improve stemness, survival, and metastatic progression[43,44]. Similarly, all CTC clusters expressed high levels of *TIMP1, JUN,*

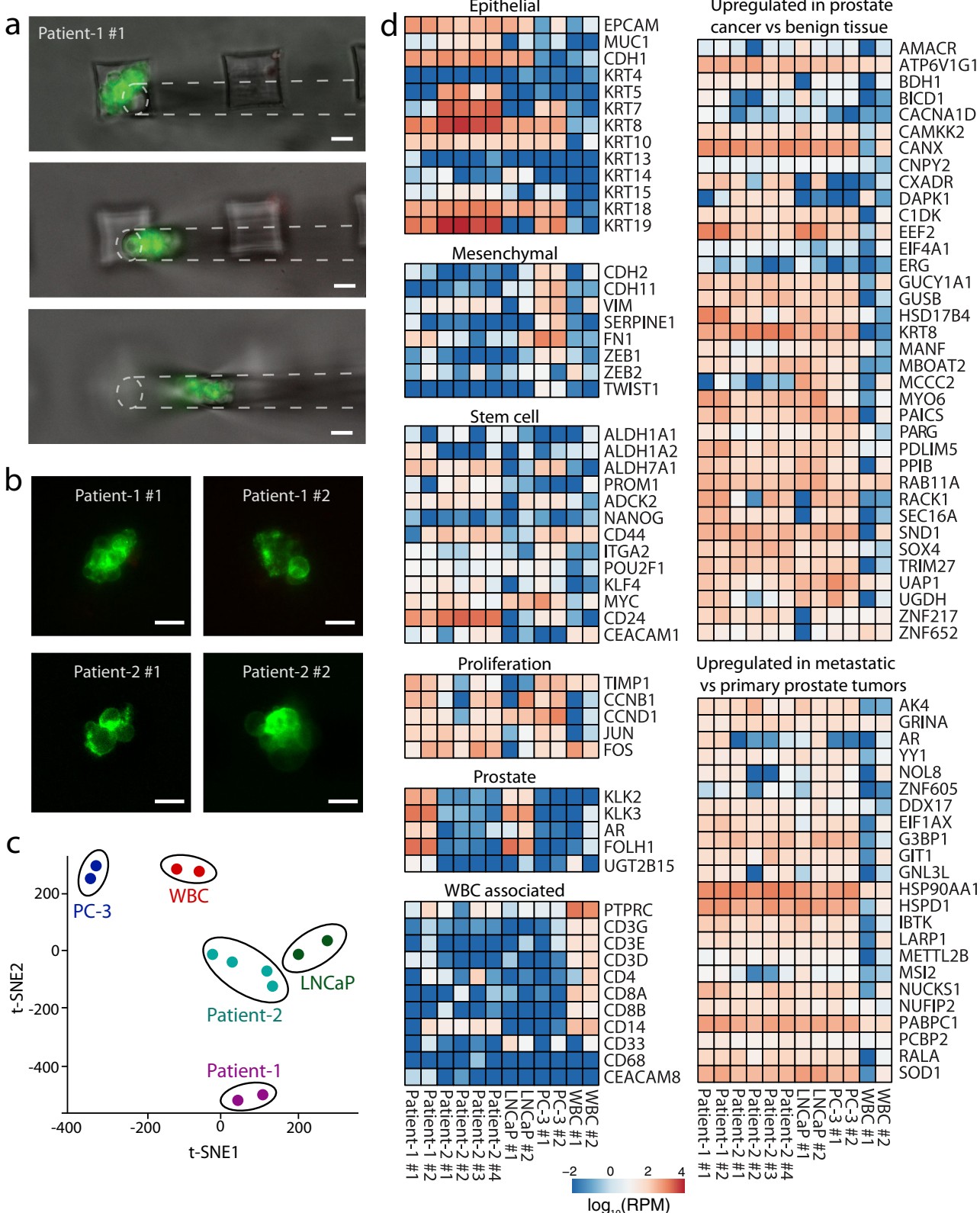

and *FOS*, which are known to stimulate cell proliferation, inhibit apoptosis, and regulate angiogenesis[45,46]. Furthermore, we have performed overrepresentation enrichment analysis for the "TOMLINS_PROSTATE_CANCER_UP" gene set from the MSigDB database at the Broad Institute, which lists the genes that are upregulated in prostate cancer compared to benign tissue[47]. As observed from the heatmap plot, all CTC clusters as

well as prostate cancer cell lines expressed high levels of these genes consistent with their prostate tumor origin. We have also performed a similar analysis for the "CHANDRAN_METAS-TASIS_TOP50_UP" gene set from the MSigDB database, which includes the genes upregulated in metastatic prostate cancer tumors compared to the primary tissue[48]. Upregulation of the genes associated with metastasis in all clusters is in agreement

**Fig. 5 Molecular analysis of the isolated CTC clusters from prostate cancer patients. a** Micromanipulation of a CTC cluster isolated from the blood sample of a metastatic prostate cancer patient (Patient-1). Following the blood filtration and PBS wash, isolated viable CTC clusters were stained on the device using FITC-conjugated antibodies against EpCAM and prostate specific membrane antigen (PSMA), scanned using fluorescence microscope, and micromanipulated directly from the device. Scale bars, 50 μm. **b** Representative fluorescence microscope images of the prostate CTC clusters that are subjected to RNA-Seq. Images were taken before they were retrieved from the device for molecular analysis. Scale bars, 20 μm. **c** t-distributed stochastic neighbor embedding (t-SNE) plot of the sequenced patient clusters, prostate cancer cell lines, and white blood cells. **d** Heatmap plot illustrating the expression levels of selected genes in isolated CTC clusters from two metastatic prostate cancer patients, prostate cancer cell lines, and white blood cells. RPM reads per million.

with the enhanced metastatic potential of CTC clusters[9,10]. Among the set, we observed the highest expression levels in *HSPD1* and *HSP90AA1* genes that are members of heat shock proteins (HSPs), playing important roles in cancer development and invasion, progression, metastasis, and drug resistance in various cancer types[49,50]. Also, all CTC clusters expressed high levels of *G3BP1* which has been correlated with the malignant degree of the tumor[51] and observed to be most abundant in castration-resistant prostate cancer (CRPC)[52], which agrees with the clinical diagnosis of both Patient-1 and Patient-2 (Supplementary Table 2).

## Discussion

In this study, we introduce a cell isolation technology that selectively detects CTC clusters in unprocessed blood samples of cancer patients. By capturing CTC clusters directly in engineered microwells etched on a polymer membrane, the Cluster-Wells offers advantages over other methods of finding these metastatic precursors in blood samples. First, the massive number of microwells screening blood constituents in parallel makes it possible to analyze clinical samples in minutes, while still ensuring the integrity of CTC clusters by subjecting them exclusively to flow speeds less than those in physiological circulation. Second, the planar device format enables the shaping of microwell traps directly in the plane of contact with CTC clusters, which removes randomness in cluster-trap interaction and allows even doublets to be captured with high sensitivity. Third, isolating CTC clusters in microwells instead of with a traditional membrane filter protects them from external stresses during or post-processing, leaving them accessible for downstream assays and eliminating the need for plating cells on a dish. Finally, the developed CTC cluster assay requires only a commercial filter holder and no additional specialized hardware other than a pump, making it readily integrable in basic research or clinical workflows.

Using the Cluster-Wells on clinical blood samples, we found CTC clusters in both prostate and ovarian cancer patients. Notably, CTC clusters were found in the peripheral blood of most of the ovarian cancer patients at relatively high concentrations. This finding is important because the spread of the ovarian cancer to surrounding organs in the peritoneal cavity is thought to be mainly mediated by cellular aggregates in the ascites, the presence of which is associated with chemo-resistant recurrent disease and poor prognosis[53,54]. While CTC clusters were found to be composed of smaller cells than tumor spheroids in ascites, they were as large as >150 cells in the samples tested, two observations seemingly at odds with each other and which have potential implications for the circulation of CTC clusters in the body. The frequency of CTC clusters in the peripheral blood of ovarian cancer patients may present valuable opportunities for further study to better understand hematogenous dissemination of ovarian cancer[55] as well as opportunities for early detection of disease onset and tumor recurrence—two major problem areas for ovarian cancer, since the disease is asymptomatic early in its progression/recurrence[56].

RNA sequencing of individual CTC clusters collected from microwells made it possible to investigate not only the inter-patient tumor heterogeneity, but also the intra-patient heterogeneity of CTC clusters and revealed features normally hidden in tissue-level analysis. All isolated prostate cancer clusters were found to strongly express epithelial markers at varying levels, while having low expression of mesenchymal markers. Each patient cluster was found to consistently express most of the genes reported to be upregulated in prostate cancer, consistent with their identity as prostate tumor origin. In addition, the upregulation of stem cell-associated markers and high expression levels of genes associated with the invasiveness of the prostate tumor cells in all isolated CTC clusters were in agreement with the previous reports on the significantly enhanced metastatic success of these cellular aggregates. Besides those common traits, comparative analysis of transcriptomes from individual CTC clusters also revealed distinct patient-specific profiles. The clusters isolated from the first patient expressed *AR*, *KLK3* (*PSA*), or *FOLH1*(*PSMA*), displaying a profile similar to androgen-responsive LNCaP cells sequenced as one of the positive controls for prostate cancer cells. In contrast, in all of the clusters isolated from the second patient, those prostate-specific genes were found to be downregulated as were in androgen-independent PC-3 cells, which were included as the other positive control prostate cancer cell type in our study.

The Cluster-Wells enables rapid screening and high-sensitivity isolation of intact CTC clusters from unprocessed whole blood samples without dependence on tumor-specific labels, thus maximizing efficient isolation of heterogeneous cancer cells. Using batches of precisely engineered cluster traps established on a polymer membrane filter, the Cluster-Wells captures and preserves even the most fragile CTC clusters for further investigation. Because the Cluster-Wells can readily be integrated to clinical or research workflows with no sample preparation, complicated technical training, or specialized instrumentation other than a pump, it is a tool that can help democratize investigations on the role of CTC clusters in cancer metastasis and also establish their clinical utility as prognostic markers for managing the disease.

## Methods

**Ethical statement.** Blood samples from consenting healthy donors were collected at Georgia Institute of Technology. The study included 7 healthy participants (5 male, 2 female, age range 24 to 37). The blood samples from prostate cancer patients were collected at Emory University Hospital and Grady Memorial Hospital, and the blood and ascites samples of ovarian cancer patients were collected at Northside Hospital. The study included 10 prostate cancer patients (male, age range 59–71) and 10 ovarian cancer patients (female, age range 27–78). All collections were performed under protocols approved by the Institutional Review Boards of Northside Hospital, Emory University, and Georgia Institute of Technology. The participants were not compensated for their enrollment.

**Microfabrication of the silicon master-mold.** The master-mold, i.e., negative pattern of the Cluster-Wells, was fabricated from a 500 μm-thick, 4-inch diameter, (100) silicon wafer using a three-mask process (Supplementary Fig. 2). The SC1813 photoresist (Shipley, Marlborough, MA) was used as the first mask. Following the spinning and patterning of the photoresist, the Si wafer was etched to a depth of 8 μm using deep reactive ion etching (DRIE) to form the square pillars. Then, a thin layer (300 nm) of low-stress nitride was deposited in an LPCVD furnace. To pattern the deposited nitride, the wafer was coated with SPR 220-7.0 positive

photoresist (Shipley, Marlborough, MA) and exposed by maskless aligner (Heidelberg MLA150). The nitride layer beneath the exposed region was etched using reactive ion etching to form a hard mask for the wet etching process. The silicon was anisotropically etched in a 45% KOH solution at 80 °C for 15 min to form the slanted walls. Finally, in order to form the negative pattern of microwells, SPR 220-7.0 photoresist was used as the mask to etch 50 μm-deep square trenches in the wafer using DRIE. Following the piranha (3:1 mixture of concentrated sulfuric acid ($H_2SO_4$) with hydrogen peroxide ($H_2O_2$)) cleaning at 120 °C for 10 min, the micromachined silicon wafer was coated with trichloro(octyl)silane under vacuum conditions for 8 h prior to PDMS casting.

**Molding of the Cluster-Wells from the silicon master**. Following the micromachining of a silicon master-mold (see Methods: molding of the Cluster-Wells from the silicon master, Supplementary Fig. 2), polydimethylsiloxane (PDMS) mold preparation and polymer device patterning were performed in a laboratory environment. First, PDMS prepolymer and its cross-linker (Sylgard 184, Dow Corning, Auburn, MI) were mixed at 10:1 ratio and poured onto the silicon master-mold. After degassing in a desiccator for an hour, PDMS was cured in an oven at 65 °C for 4 h, then cut and peeled-off from the silicon master-mold (Supplementary Fig. 3a, i). The surface of the PDMS layer was activated using oxygen plasma and coated with trichloro(octyl)silane for 8 h. Previously fabricated and silane coated PDMS layer was used as a mold for the fabrication of the secondary PDMS mold, having the negative pattern of the device (Supplementary Fig. 3a, ii). Following the peel-off (Supplementary Fig. 3a, iii), secondary PDMS mold was used for patterning the photocurable perfluoropolyether (PFPE) based polymer (Fluorolink MD700, Solvay), which is used as the device material. Following the PDMS-to-PDMS molding process, the secondary PDMS mold (Supplementary Fig. 3b, i) was placed on a polyethylene terephthalate (PET) sheet and filled with Fluorolink MD700 mixed with 4% w/w 2-hydroxy-2-methylpropiophenone (Darocur 1173) under 50 mbar vacuum. Once the mold is filled, polymer is cured with UV light (Fig. 3a, iv), the secondary PDMS mold was peeled-off from the PET sheet/device stack (Supplementary Fig. 3a, v), and the device (Supplementary Fig. 3b, ii-iv) was released from the underlying PET sheet on a thermoelectric cooler at 4 °C, which promotes easier release of the polymer device without any damage (Supplementary Fig. 3a, vi).

**Measurement of the device sensitivity**. For the characterization of the Cluster-Wells, we utilized analytical versions of the devices having smaller effective filtration area. While using a range of flow rates from 1 to 1.5 mL/h due to camera's recording frame rate limitation, the flow speeds at micromeshes were arranged by varying the size of the devices, where decrease in the size corresponded to increase in the flow rate. In these characterization studies, the volume of blood sample was also adjusted, within practical limits, to ensure the number of blood cells processed by each microwell was similar to that of the actual device (with 47 mm diameter) when processing 10 mL of whole blood for each tested condition. Consequently, we used ~750, ~375, and ~200 μL of whole blood to test the device performance for flow rates of 25, 100, and >250 mL/h, respectively. Smaller diameter devices were mounted to a commercially available 13 mm diameter filter holder (GE Healthcare Life Sciences) with a size matching hollow PDMS layers, which limit the fluid flow to the region of interest with a certain diameter and prevent stagnant flow region formation inside the filter holder. Before introducing the sample, the experimental setup (Fig. 2b) was primed with pure ethanol, which was followed by a PBS wash. Then, the setup was incubated with 3% bovine serum albumin (BSA) for 1 h to minimize non-specific cell adhesion. The Hoechst dye stained cells (see separate section on cell culture and preparation) were spiked into whole blood and run through the experimental setup using a syringe pump (Harvard Apparatus Infuse/Withdraw PHD Ultra) at the withdrawal mode. The blood with spiked cell clusters traversed through input channel (50 μm in height, 500 μm in width) of 2-channel microfluidic interface, device/filter holder assembly, and lastly, output channel (50 μm in height, 500 μm in width) of the microfluidic interface. The input and output microfluidic channels were simultaneously video recorded at 50 frames per second in fluorescent (DAPI) channel using an inverted fluorescence microscope (Eclipse Ti, Nikon, Melville, NY) for tracking the cells entering and exiting the analytical version of the Cluster-Wells, respectively. Then, the recorded video was processed by a custom-built software (Visual Studio, 2017), and the count of clusters entering and leaving the device was obtained with respect to number of cells within each cluster, which is used for the calculation of the capture efficiency.

To test the reliability of the imaging system, we operated the 2-channel microfluidic interface in a loop without the device attached, where outlet of the input channel was connected to the inlet of the output channel using an 8 cm long 0.02" ID tubing. The experiment was performed using a syringe pump set at 1.5 mL/h and both microfluidic channels were video recorded at 50 frames per second using fluorescence (DAPI) channel. Custom written software was used to extract the frames having cell events in either of the microfluidic channels. After analyzing the frames, we observed that there was virtually no loss in the system when the size distribution of the clusters passing through input and output channels were compared (Supplementary Table 1). To assess the validity of the data obtained using microfluidic setup, we performed capture efficiency experiments at 25 mL/h by directly counting the isolated clusters on the device. In these experiments, the spiked population was imaged using a microfluidic channel as described earlier to have exact spiked cluster counts. The clusters were injected into an EDTA tube having 10 mL of whole blood and the spiked blood sample was subsequently processed at 25 mL/h using Cluster-Wells. The device was imaged using fluorescence microscope and the captured cluster counts were compared with the spiked population for measuring the capture efficiency (Supplementary Fig. 4), which matched with the capture efficiency data obtained using microfluidic interface.

**Cell culture and preparation**. As model biological samples, we used HeyA8 (obtained from Dr. John F. McDonald, Georgia Institute of Technology), LNCaP (ATCC-CRL-1740; Manassas, VA), MDA-MB-231 (ATCC-HTB-26; Manassas, VA) and MCF-7 (ATCC-HTB-22; Manassas, VA) cell lines. The cell lines were cultured in RPMI-1640 (LNCaP and HeyA8) or DMEM (MDA-MB-231 and MCF-7) medium containing 10% fetal bovine serum (FBS) (Seradigm, Radnor, PA) in 5% $CO_2$ atmosphere at 37 °C. Once they reach 80% confluence, cells were detached from the culture flask using 0.25% trypsin (Gibco) for 2 min. Subsequently, cells were pelleted, the supernatant was removed, and the cells were resuspended in 1× PBS solution by gentle pipetting. For all cell lines used in this study, pelleted cells were initially found to adhere to each other in groups of hundreds to thousands of cells and were subsequently dissociated into smaller clusters within the size range of interest for our study mechanically by pipetting.

For the capture efficiency experiments, nuclear staining was performed for visual tracking of clusters. Before detaching cells from the flask surface, nuclei of cells were labeled by incubating the cells in a 4 mL Hoechst 33342 (Thermo Fisher – Cat No: H3570) dye solution (1:1000) for 20 min in 5% $CO_2$ atmosphere at 37 °C. After washing off the staining solution, the cells were detached and resuspended in 1× PBS solution. Small aliquots (<10 μL) of the suspension were used to prepare the test samples containing a few hundred (184–552 clusters) cell clusters per experiment. The number of spiked clusters in our experiments was set to ensure (1) the spiked cluster population was large enough to contain different-sized clusters for a comprehensive test and (2) the number of clusters did not saturate the device with the majority of the wells remaining empty.

**Measurement of the cell viability**. To determine the viability of cells, live/dead assay (ab115347, Abcam, Cambridge, MA) was performed according to the manufacturer's instructions. The assay was mixed at 5× concentration (1:200) with the 1× PBS solution containing the control and released cells. Following the incubation in dark environment for 15 min, the samples were imaged with an inverted fluorescence microscope (Eclipse Ti, Nikon, Melville, NY), where the live and dead cells were observed in green (FITC) and red (TexasRed) channels, respectively (Supplementary Fig. 9).

**SEM sample preparation and imaging**. The captured clusters were first fixed in 2.5% glutaraldehyde and then in 1% osmium tetroxide, both diluted in 0.1 M sodium cacodylate. After fixation, cells were dehydrated in 50%, 70%, 80, and 95% ethanol solutions in water and 100% ethanol successively for 15 min in each. The sample was washed with hexamethyldisilazane (HMDS) (Electron Microscopy Sciences, Hatfield, PA) and dried at room temperature overnight. The device and captured clusters were coated with Pt/Pd using a sputtering system and imaged using a Hitachi SU8230 scanning electron microscope.

**Sample collection**. Blood samples from consenting healthy donors were collected at Georgia Institute of Technology under an IRB-approved protocol. The blood samples from prostate cancer patients were collected at Emory University Hospital and Grady Memorial Hospital, and the blood and ascites samples of ovarian cancer patients were collected at Northside Hospital under IRB-approved protocols. The blood samples were collected in EDTA tubes (BD Vacutainer) and processed within 4 h of blood draw. To prevent sedimentation, tubes were placed on a rocker until use. Ascites samples from ovarian cancer patients were transferred to Georgia Institute of Technology in evacuated glass containers and processed within 4 h of withdrawal.

**Sample processing**. The devices were placed inside the filter holders and wetted with pure ethanol. Following the wetting, ethanol was washed away with 1× PBS. Prior to use, the device/filter holder assembly was incubated with 3% BSA for 1 h to minimize non-specific cell adhesion on the surfaces. Then, the BSA was washed away with 1× PBS before the introduction of blood or ascites samples. Both ascites and unprocessed whole blood samples were run through the devices using a syringe pump (Harvard Apparatus Infuse/Withdraw PHD Ultra) under withdrawal mode at 25 mL/h flow rate. Then, the devices were washed using 1× PBS solution for 1 h before the immunofluorescence staining (see separate section on immunofluorescence staining of CTC clusters).

**Immunofluorescence staining of CTC clusters**. After processing the samples, the captured cells were fixed with 4% paraformaldehyde (Electron Microscopy Sciences, Hatfield, PA) for 10 min, and subsequently permeabilized with 1% Triton-X (Sigma-Aldrich, St. Louis, MO) in PBS for 10 min. Prior to immunofluorescence staining, the device/filter holder assembly was incubated with a blocking buffer containing 2% goat serum and 3% BSA for 30 min. For prostate samples, Cytokeratin 8/18 (Invitrogen - Cat No: MA5-32118, Clone: SU0338, (1:400)), Vimentin (Invitrogen - Cat No: MA5-14564, Clone: SP20, (1:1000)), PSA/KLK3 (Cell Signaling Technology - Cat No:

5365S, Clone: D6B1, (1:750)), EpCAM (Invitrogen - Cat No: MA5-29246, Clone: 28, (1:400)) and Anti-CD45 (BD Biosciences - Cat No: 555480, Clone: HI30, (1:500)), and for ovarian samples, Cytokeratin 7 (Invitrogen - Cat No: MA5-32173, Clone: ST50-05, (1:400)), Cytokeratin 8/18 (Invitrogen - Cat No: MA5-32118, Clone: SU0338, (1:400)), Vimentin (Invitrogen - Cat No: MA5-14564, Clone: SP20, (1:1000)) and Anti-CD45 (BD Biosciences - Cat No: 555480, Clone: HI30, (1:500)) primary antibody cocktail was introduced and incubated overnight. The excess antibodies were washed away with 1× PBS. Then, the matching secondary antibodies Alexa Fluor 488 (Invitrogen - Cat No: A-11008, (1:500)) and Alexa Fluor 594 (Invitrogen - Cat No: A21125, (1:500)) were run through the device, incubated for 1 h, and washed away with 1× PBS. The nuclei of cells were stained with 4′,6-diamidino-2-phenylindole (DAPI) (Invitrogen - Cat No: D1306, (1:1000)) for 10 min, and the devices were washed with 1× PBS. Lastly, devices were removed from the filter holders and mounted between two glass slides for imaging.

**Micromanipulation of CTC clusters from patient samples.** The captured, alive CTC clusters were stained with Alexa 488-conjugated antibodies against EpCAM (Cell Signaling Technology – Cat No: 5198S, (1:400)), prostate-specific membrane antigen (PSMA) (Biolegend – Cat No: 342506, Clone: LNI-17, (1:80)), and the contaminating WBCs were stained with PE-CD45 (TRITC) (BioLegend – Cat No: 368510, Clone: 2D1, (1:40)). The device with captured clusters was mounted in a Petri dish, and then imaged using inverted fluorescence microscope (Eclipse Ti, Nikon, Melville, NY). Identified clusters were directly micromanipulated from the device and transferred to the PCR tubes containing RLT (Qiagen) + BME (Sigma-Aldrich) buffer using Eppendorf TransferMan 4r micromanipulator. During micromanipulation, the dish was supplemented with 1× PBS to prevent microwells from drying, which was observed to take ~10 min at room temperature, Tubes were vortexed for 1 min and transferred to −80 °C freezer. Most of contaminating WBCs were observed to remain adhered to the device during cluster retrieval. Nevertheless, CTC clusters micromanipulated from Cluster-Wells were discharged into an empty Petri dish and were then repicked to ensure against potential artifacts from WBC contamination in RNA sequencing.

**Viability assessment of CTC clusters isolated from patient samples.** Blood sample was processed with the Cluster-Wells at a flow rate of 25 mL/h. The nuclei of the captured cells were stained with Hoechst dye (Thermo Fisher – Cat No: H3570, (1:1000)), and the membranes of the contaminating WBCs were stained with PE-CD45 (TRITC) (BioLegend – Cat No: 368510, (1:40)) for negative identification of potential CTC clusters on the device. Enriched CTC clusters were first imaged on the device as the baseline (Supplementary Fig. 11, i) and CTC clusters on the device were subsequently subjected to membrane integrity-based viability assay, NucRed Dead 647 Ready Probes (Thermo Fisher – Cat No: R37113), per manufacturer-provided protocol. Fluorescence images of the assayed CTC clusters were taken to record the viability reporter dye fluorescence (Cy-5) (Supplementary Fig. 11, ii). To prove their tumor origin, cells were then fixed with 4% PFA, device was placed inside a filter holder, and fixed staining protocol was applied using antibodies against Cytokeratin 7, 8/18, Vimentin (FITC), and CD45 (Texas Red) and clusters were imaged with a fluorescence microscope (Supplementary Fig. 11, iii). The viability assay was then validated by assaying the fixed CTC clusters as a negative control and the CTC clusters were imaged once more with a fluorescence microscope to confirm a negative viability result due to compromised membrane of the fixed cells (Supplementary Fig. 11, iv).

**RNA library preparation and sequencing.** RNA was extracted from the lysed samples using the Macherey-Nagel Nucleospin RNA XS kit. Double stranded cDNA was generated using the Takara Bio SMART-Seq v4 ultra low input RNA kit for sequencing. Barcoded Illumina compatible sequencing libraries were then generated using the Illumina Nextera XT DNA library preparation kit with a reduced tagmentation time of 3 min and the samples were cleaned using a 1:1 bead ratio to eliminate the presence of primer dimers. The qualities of the libraries were determined on the Agilent Bioanalyzer using a DNA high sensitivity chip and the concentrations were determined using the Invitrogen Qubit fluorometer. The barcoded samples were then pooled and sequenced on the Illumina NextSeq 500 and NovaSeq 6000.

**RNA sequencing data analysis.** Raw fastq files were analyzed for quality using FASTQC and trimmed using TrimGalore (ref: https://github.com/FelixKrueger/TrimGalore) to remove adapter sequences and low quality reads (minimum Phred score >24). Trimmed reads were mapped to the hg38 build of the human genome using STAR mapper (PMID: 23104886) and transcripts were quantified by mapping to the GenCODE.v24 annotation version of the human transcriptome. For these prostate CTC clusters, a median of 83.15 M reads were input to STAR mapper (range 53.7–129.79 M), and a median of 88.8% reads mapped uniquely to the human transcriptome (range 80.57–91.59%). A total of 16,412 transcripts were detected with at least 10 mapped reads in one sample and used for DESeq2 analysis in R (PMID: 25516281). A total of 12,149 mapped Ensembl genes ranked by DESeq2 log2foldchange were used as input for GSEA analysis (PMID: 17644558), using the WebGestalt tool (PMID: 28472511). Enriched gene sets were analyzed using the Cancer Hallmark 50 gene sets and the KEGG Pathway gene sets. t-SNE analysis was performed in R using the M3C package with seed = 123 and

perplexity = 1. Total read counts were normalized to FPKM values using Cufflinks (PMID: 22383036). A total of 26,391 transcripts had an FPKM > 1 in at least one sample, and 10,885 transcripts had a median FPKM > 1. To be used in the heatmap plot, reads per million (RPM) count for the genes was generated. Lastly, the heatmap plot was generated using ClustVis online tool.

**Statistics and reproducibility.** Characterization experiments were performed as three independent experiments. The data are presented as mean ± SD unless otherwise stated. No statistical method was used to predetermine sample size. The experiments were not randomized. The fabricated devices were chosen randomly for the characterization experiments and patient sample processing. Patient samples were allocated into different groups based on their tumor origin (prostate vs ovarian). Blinding was used for processing the patient blood samples, where all patient information other than cancer type was withheld until the end of the study. RNA sequencing data was processed by unbiased bioinformatics pipelines prior to evaluation by the authors. The representative experiments shown in Figs. 1b, 2a, 3e and Supplementary Figs. 3b, 9, 10 were performed once for illustration purposes. Similarly, Figs. 4a–c, 5a, b, and Supplementary Figs. 10, 11a, b correspond to specific patient samples and experiments were performed once at the time of processing these samples.

**Reporting summary.** Further information on research design is available in the Nature Research Reporting Summary linked to this article.

## Data availability

The authors declare that the data supporting the findings of this study are available within the paper and its Supplementary Information file. The raw and processed sequencing data generated in this study have been deposited in the NCBI's Gene Expression Omnibus database (GEO) under accession code GSE202650. Publicly available datasets used in this study are Hallmark50 gene sets [https://www.gsea-msigdb.org/gsea/msigdb/genesets.jsp?collection=H], hg38 build of the human genome [https://www.ncbi.nlm.nih.gov/assembly/GCF_000001405.40], KEGG pathway database [https://www.genome.jp/kegg/pathway.html], "CHANDRAN_METASTASIS_TOP50_UP" gene set [https://www.gsea-msigdb.org/gsea/msigdb/cards/CHANDRAN_METASTASIS_TOP50_UP], "TOMLINS_PROSTATE_CANCER_UP" gene set [https://www.gsea-msigdb.org/gsea/msigdb/cards/TOMLINS_PROSTATE_CANCER_UP]. Source data are provided with this paper.

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

## Acknowledgements

The authors would like to thank all patients and healthy blood donors who participated in this study. The authors also thank Dr. Anton V. Bryksin and Naima Djeddar from Georgia Tech Molecular Evolution Core, Sakeenah Hicks, Dr. Christopher Scharer and Dr. Lyra M. Griffiths from Emory Integrated Genomics Core (EIGC), and Dr. Henry R. Johnston and Dr. Robert A. Arthur from Emory Integrated Computational Core (EICC) for their contributions in RNA-sequencing and data analysis; and staff at Georgia Tech Stamps Health Center for their help with the blood drawal from healthy donors. This work was supported by the start-up funds provided to A.F.S. by Georgia Institute of Technology. The research was also supported in part by the Office of the Assistant Secretary of Defense for Health Affairs through the Prostate Cancer Research Program under Award No. W81XWH-20-1-0649 (A.F.S.), Georgia Tech Petit Institute Seed Grant (A.F.S. and J.F.M.), the Dunwoody Golf Club Prostate Cancer Research Award, and Winship Invest$ Pilot Grant from the Winship Cancer Institute of Emory University (A.F.S. and M.A.B.).

## Author contributions

M.B. and A.F.S. designed the research and wrote the manuscript. M.B. and R.L. fabricated the devices. M.B. conducted the experiments. M.B. and N.A. developed the analysis software. M.B. and A.F.S. analyzed the data. S.T., L.D.M., B.N., O.K., M.G.S., B.B.B., M.A.B., and J.F.M. helped with the acquisition of patient samples. M.B., T.O.A., B.E.S., and C.H.C. processed and analyzed the patient samples. S.B. helped with the library preparation and RNA sequencing. C.S.M. processed and analyzed the sequencing data. T.O.A., N.A., O.C., and D.L. contributed to the manuscript preparation. All authors read and approved the final manuscript.

## Competing interests

M.B. and A.F.S. are inventors on a patent Georgia Institute of Technology filed (US patent application number 17/619,276). The patent application covers the developed CTC cluster isolation technology. The remaining authors declare no competing interests.
