## [Peer Review File · Nature Communications]

Reviewers' Comments:

Reviewer #1:

Remarks to the Author:

In this article, Boya et al. present a new microfluidic chip specifically designed to enrich CTC clusters.

This new device may attract attention since it focuses on cluster enrichment and CTC clusters are even more rare events than single CTC but present probably highly metastatic potential. Indeed, among all new technologies developed to enrich CTC, most of them focus on enriching single CTC but the accent is not put on enriching specifically CTC clusters. This new device is also attractive by the high speed it processes the blood (up to 25ml in one hour), which guarantees viability and nice quality of some tricky downstream analysis like RNA sequencing and by the simple equipment it requires which makes it readily and easily implementable in other labs.

The paper is well written, in a very understandable way, notably also for the technical parts for a non-specialized audience.

However, there are few points that can be addressed to improve the quality of the manuscript:

(1) The technical parts are nicely described but the part dedicated to biology is very short. The authors provide the proof of principle on very small cohort of patients (N=9 ovarian cancer and N=7 prostate cancer patients). The authors could also have detailed more the results on the number of cells present in each clusters (giving the median for example, line 214) and also how many were pure CTC clusters or hybrid ones, containing leukocytes.

(2) Moreover, RNA sequencing that was performed on clusters is "only" made on 2 patients and does not reveal particularly new biological insights that were not previously described. However, on a more positive way, it is already satisfying to confirm, on completely separated patients, previous biological findings.

(3) Better description of the quality of RNA sequencing could also be provided to really show a potential advantage of a rapid enrichment.

(4) Further information on the count of single CTC by a standard device would be helpful. This might have also illustrated that current classical CTC enrichment devices can miss CTC clusters.

(5) Finally, the authors should also precise the volume of blood that was processed for each patient. There is no side to side comparison with the previous microfluidic chip dedicated to CTC clusters on patients samples.

(6) On the technical parts, it would be really important to present the volume of blood that was processed and the number of replicates in the recovery assays (starting from line 127 and figure 2b). The recovery rates are calculated in an indirect manner and one could wonder if it would have been possible to count directly on the chip.

(7) A validation on healthy donors should also have been made to show assay specificity.

(8) Regarding the RNA analysis of the cell clusters, there is no comparison with normal cells (leukocytes) and tumor cell line cells to proof that the RNA profiles they obtained are actual tumor cells. Better would have been whole genome sequencing.

(9) I would also like to see a few more experiments to proof that the patient cell clusters are really viable (e.g., short term CTC culture).

MINOR POINTS:

Figure 2f: The absolute number of leukocytes (i.e., not in percentage) should be also described in the text.

In the material and methods section, the name of the antibody clones should be given.

Lines 113-115 and reference to fig 2a: it is said in the manuscript text that cells were pre-labelled with fluorescent dyes and Fig2a shows a CK staining. These lines should be revised.

Lines 200-202: the authors should precise how long the procedure takes.

Reviewer #2:

Remarks to the Author:

Overall, I believe the authors have put together a comprehensive study on a unique microfilter device for CTC cluster isolation. While the scope of the paper is rather conservative, the authors address an important niche in the high throughput isolation of CTC clusters. The novelty of the microfilter design is profoundly unique from previous cluster isolation devices, however I do believe that other devices – particularly the continuous isolation microfluidic device developed by

Ebb et al earlier this year – have achieved similar isolation efficiencies at comparable throughputs. That being said, the ease of operation of this microfilter combined with the downstream compatibility with single-cluster molecular analytics could potentially give this device a competitive edge and impact the field in a meaningful way.

The authors outline the fabrication and characterization of polymer microfilter able to isolate CTC clusters from multiple cancer types at high throughputs. The microfilter uses a combination of silicon micromachining, soft lithography, and micro molding techniques, resulting in an efficient, cheap, and easily disposable plastic device. Notably, the filter is made of a photocuring, low-bind, fluorine-based polymer to reduce non-specific reactions. The microwell design is shown to have very high capture efficiency for CTC clusters comprised of greater than 2 CTCs per cluster. The wells are also shown to have a high cluster release efficiency and are easily accessible for individual micromanipulation of clusters. The authors validate their device functionality in both cell lines and patient samples. The authors also validate the downstream applicability of their microfilter by individually selecting 7 CTC clusters from two different prostate cancer patients to perform RNA sequencing. This analysis revealed both inter- and intra-patient heterogeneity in the gene expressions of the CTC clusters.

The authors do an excellent job of creating a device to specifically tackle the issue of CTC cluster isolation at a clinically relevant throughput. As established in the study, this microfilter provides a cheap, reproducible way to study CTC clusters and their biological relevancy in metastasis. The compatibility of the microfilter with micromanipulation and – by extension – single-cell RNA sequencing offers a very attractive mode to study cluster biology. However the authors should address the following minor concerns

1. The authors should discuss their device in the context of the other technologies, specifically the continuous flow ones

The novelty of this isolation device is based on the devices 1) throughput and 2) gentle flow conditions (i.e. the clusters experience shear stresses less than those found under physiological flow conditions). While the authors do an excellent job in showing the advantages of their device over the Cluster-Chip device (ref. 23), they only briefly discuss the benefits of their microfilter design over the continuous microfluidic device referenced in ref. 29. The authors claim the disadvantage of the of the continuous isolation device is that the higher flow rates threaten to damage cluster integrity (line 70-71). The authors of this study (Edds, J. F. et al), however, report a minimal loss in cluster integrity from their device. While the design of the two devices is very different – one is a continuous, PDMS microfluidic device while the other is a microfabricated filter – they both claim a gentle recovery of CTC clusters under similar throughput.

2. The authors do an excellent job in characterizing the capture and release efficiencies of their microfilter device. The authors also sufficiently address the viability of their released clusters from the devices following processing. One additional piece of evidence that would be helpful in supporting their claim is some kind of figure or statistic showing that the integrity of their clusters does not change between capture and release. The authors do an excellent job explaining the theory behind the advantages of flowing the wells in parallel, but they do not provide any data that I could find on how the distribution of cluster sizes changes from the start of processing, to capture and then release. This study would help to reinforce their claim that the filter operates under truly gentle conditions. Additionally, a simple simulation or mathematical explanation demonstrating the effective flow rate or shear stress through each well would be a helpful supplement to explanation.

3. One additional piece of information that may be insightful would be to include patient treatment plan in the patient information table in the supplemental information. From the information provided, it is unclear at what point in the patient's treatment the blood is processed: are the patients treatment naïve, under treatment, or some combination of the two? Are any of the samples follow up samples? This information may help to provide further insight. For example, there seems to be a slight inverse correlation between CA125 levels and cluster abundance in ovarian patients; understanding the patients disease status and treatment path may help to explain potential trends.

Sunitha Nagrath

Reviewer #3:

Remarks to the Author:

In the manuscript titled "High throughput, label-free isolation of circulating tumor cell clusters in meshed microwells", Boya et al developed a microfabricated filtration chip for capturing rare CTC clusters. They perform a thorough validation of the system using several cell lines, and then show the ability to detect clusters from several patient samples, concluding with the RNA sequencing of 7 clusters from two patients with prostate cancer. The device design is a novel strategy for isolating CTC clusters, and they present data showing a 10-fold increase in effectiveness compared to a leading technology (Cluster-chip).

The strengths of this manuscript include using a novel fabrication process to make the mesh microwells for capturing CTC clusters and providing extensive characterization with both cell lines and detection of clusters in blood from patients with metastatic ovarian and prostate cancer. Although the authors provide sequencing of 7 individual clusters, there does not seem to be clear findings that emerge. Overall the manuscript validates the superiority of their technology to previous approaches but does not show how it can be exploited to reveal novel biological or clinical insights. Nevertheless, I recommend the manuscript is accepted provided the comments below are addressed.

Major concerns:

1. The authors should include the published recovery rate and retrieval rate of Cluster-chip.
2. The authors should provide the distribution (and median size) of the clusters.
3. The claim that ascites clusters have larger cells has not been substantiated. The ascites staining is for EpCAM, a surface marker that stains the membrane and allows for clear size measurement. However, the CTC cluster has a staining for vimentin and cytokeratin, both of which are intercellular proteins and cannot be used by fluorescent imaging to measure size. A brightfield comparison or surface stain (e.g. EpCAM) would allow for a direct comparison of size of the cells. More data than the single image shown should be provided (e.g. measurements of the cells or multiple images). The number/concentration of clusters found in ascites should be included.
4. The presentation of the sequencing data is not entirely clear. For one patient, (PCa-2) none of the 5 clusters sequenced express any of the prostate markers (including the antibody used to identify the clusters- FOLH1/PSMA), and very few of the epithelial (and even mesenchymal markers). And the claim "PCa-2 CTC clusters presented similar levels of expression for epithelial and mesenchymal markers" is a little misleading, since the cells do not really express either class of genes. Non-specific binding of the antibody should be confirmed. What genes are the PCa-2 clusters high in?
5. How specific are the "metastasis" genes associated with cancer, as opposed to other biological processes? These are definitely genes related to cell cycle, but it is unclear where this list of genes came from (it is not found in either of the papers cited for the sentence [line 268- citations 9 and 13]). Also, why were only 4 genes selected to define "metastasis"?
6. In the discussion, there is again an overstatement of the "hybrid" epithelial/mesenchymal expression, when in fact there is quite low expression of both classes except for Vim and KRT10. In fact, one could argue that PCa-1 has high expression for both, while PCa-2 has low expression for both E/M.
7. The statement "Furthermore, the lack of expression of AR, KLK3 (PSA), or PSMA in patient PCa-2 suggests that this patient could suffer from advanced, AR-negative prostate cancer" is entirely speculation, and is inconsistent with the fact that the sample stained positive for PSMA. A broader metastasis geneset would be helpful to confirm the nature of these clusters

Additional comments that should be addressed:

1. Figure 2- c, d, and e do not have error bars. Were there any replicates?
2. To calculate retention rate of clusters, imaging was used of pre and post mesh blood. Is there a way to validate that clusters are not being broken up by the mesh? Or that any of the cells are sticking to the device (given the very low flow rate this could arise)? Also, how accurate was the image processing system (ie, if you run just in a loop without the chip, do you still detect the same numbers on both sides)?

3. Was there any evidence of large clusters breaking up into smaller clusters?
4. All of the cell lines used appear to be adherent cell lines (ie, cell lines that do not naturally form in clusters). So "clusters" would just be different degrees of trypsinized cells? Would a cluster-forming suspension line produce different results? Also, what is the rough number of total detected clusters in the cell lines? Is this 1 cluster/mL (similar to patient data) or 10000 clusters/mL?
5. A claim made that only at 750ul/min was dissociation seen (more single cells in waste than in input). This should be substantiated with data.
6. The authors claim that there was a .05-.1% retention of WBCs in their samples. For a ~10mL sample, this would likely come out to ~50k-100k WBCs sticking in the chip (up to almost 1 per well). Where did the WBCs non-specifically bind (on the top region or in the wells)? Did WBCs come off during cluster retrieval? And would this make sequencing analysis difficult? And was there any platelet non-specific binding?
7. For calculating retrieval rate, how did the total number (both retrieved and stuck clusters) compare to the expected value (as defined by your retention rate- from imaging the pre and post blood)?
8. Was viability of the true CTC clusters assessed? It is less surprising that a robust cell line would not be damaged by the processing
9. How long did it take for the wells to dry out during the micromanipulation?
10. Line 234 claims 3/5 prostate cancer patients had clusters, but supplemental figure 1 appears to show that 5/7 patients had clusters
11. A statement is made that "the developed CTC cluster assay requires only a commercial filter holder and no additional specialized hardware or training," but it does require a syringe pump to operate as well (stated in the methods)

Dear Editor,

We are submitting our response to the Reviewers' comments on our manuscript entitled "High throughput, label-free isolation of circulating tumor cell clusters in meshed microwells". We would like to thank the Reviewers for their insightful and constructive comments, which helped us immensely in revising our manuscript.

The Reviewers indicated two major concerns to be addressed: 1) potential disruption of the clusters during capture and/or release processes with our technology and 2) the need for more comprehensive analysis of patient CTC clusters, particularly in regards to the previously presented RNA sequencing results. With respect to the first point, we have performed an extensive set of new experiments whose results are now reported in the revised manuscript. The new experiments clearly showed that the Cluster-Wells does not dissociate or damage the clusters both during capture and release under nominal operational conditions. With respect to the second point, we now present a more comprehensive molecular analysis of patient CTC clusters isolated using our technology. With the inclusion of these new samples with appropriate positive and negative controls, we could confirm the tumor origin of isolated CTC clusters from our RNA sequencing results effectively validating the applicability of our technology for downstream transcriptome analysis of CTC clusters. Below are our detailed responses to all the concerns raised by the Reviewers.

Reviewer #1:

In this article, Boya et al. present a new microfluidic chip specifically designed to enrich CTC clusters.

This new device may attract attention since it focuses on cluster enrichment and CTC clusters are even more rare events than single CTC but present probably highly metastatic potential. Indeed, among all new technologies developed to enrich CTC, most of them focus on enriching single CTC but the accent is not put on enriching specifically CTC clusters. This new device is also attractive by the high speed it processes the blood (up to 25ml in one hour), which guarantees viability and nice quality of some tricky downstream analysis like RNA sequencing and by the simple equipment it requires which makes it readily and easily implementable in other labs. The paper is well written, in a very understandable way, notably also for the technical parts for a non-specialized audience.

However, there are few points that can be addressed to improve the quality of the manuscript:

We thank the Reviewer for their favorable comments and positive outlook for our manuscript.

(1) The technical parts are nicely described but the part dedicated to biology is very short. The authors provide the proof of principle on very small cohort of patients (N=9 ovarian cancer and N=7 prostate cancer patients). The authors could also have detailed more the results on the number of cells present in each clusters (giving the median for example, line 214) and also how many were pure CTC clusters or hybrid ones, containing leukocytes.

Following the Reviewer's suggestion, we have expanded the section of our manuscript dedicated to biological results. In the revised manuscript, we present a more comprehensive analysis of the

data on clinical samples, including the number of cells present in each CTC cluster [see p. 9, ¶ 2, lines 5-7; Fig. 4f] and the percentage of pure and hybrid CTC clusters [see p. 9, ¶ 2, lines 7-9; Fig. 4g]. We also expanded our analysis of the molecular data from RNA sequencing of CTC clusters [see “Molecular analysis of patient CTC clusters” section; Fig. 5d; Supplementary Fig. 12, 13 & 14].

(2) Moreover, RNA sequencing that was performed on clusters is “only” made on 2 patients and does not reveal particularly new biological insights that were not previously described. However, on a more positive way, it is already satisfying to confirm, on completely separated patients, previous biological findings.

In the revised manuscript, we expanded the section of the manuscript dedicated to the molecular analysis in order to make the study more comprehensive as suggested by the Reviewer [see “Molecular analysis of patient CTC clusters” section; Fig. 5d; Supplementary Fig. 12, 13 & 14].

(3) Better description of the quality of RNA sequencing could also be provided to really show a potential advantage of a rapid enrichment.

We agree with the Reviewer. To quantitatively describe the quality of RNA sequencing in our study, we have provided additional metrics from the quality control of the RNA sequencing data obtained from CTC clusters [see p. 10, ¶ 3, lines 3-5; p. 20, ¶ 3, lines 5-7]. These results showed that we could reliably obtain high quality data from RNA sequencing of CTC clusters isolated using our technology.

(4) Further information on the count of single CTC by a standard device would be helpful. This might have also illustrated that current classical CTC enrichment devices can miss CTC clusters.

In our study, we did not analyze the patient samples with another device for isolation of single CTCs. However, to address the Reviewer’s comment, we cited a study comparing the performance of a cluster-optimized microfluidic device (Cluster-Chip) against 1) a standard filter and 2) an immunoaffinity-based microfluidic device optimized for capturing single CTCs. Those previously reported results illustrated the higher cluster capture sensitivity provided by a cluster-optimized enrichment process over classical CTC enrichment methods [see p. 5, ¶ 3, lines 9-10].

(5) Finally, the authors should also precise the volume of blood that was processed for each patient. There is no side to side comparison with the previous microfluidic chip dedicated to CTC clusters on patients samples.

Based on the Reviewer’s comment, we reported the volume of samples collected from each patient in our study [see Supplementary Table 2 & 3]. While acknowledging we analyzed two different patient cohorts, we have discussed CTC cluster statistics (i.e., the number of CTC clusters observed per mL of blood sample) obtained using Cluster-Wells relative to the published results from the Cluster-Chip on prostate cancer patients as suggested by the Reviewer [see p. 9, ¶ 2, lines 11-14].

(6) On the technical parts, it would be really important to present the volume of blood that was processed and the number of replicates in the recovery assays (starting from line 127 and figure

2b). The recovery rates are calculated in an indirect manner and one could wonder if it would have been possible to count directly on the chip.

We agree with the Reviewer. In the revised manuscript, we included information on the volume of blood processed in our characterization experiments [see p. 15, ¶ 2, lines 5-9]. We also included the number of replicate experiments performed in figure captions as suggested [see Fig. 2c, d, e].

Based on Reviewer's suggestion, we also performed new experiments to measure the device capture efficiency, this time by counting the captured cells directly on the device and showed the measured capture rates from on-device cluster count matched with the rate calculated by counting the spiked and missed clusters [see p. 5, ¶ 1, lines 8-10; p. 16, ¶ 1, lines 8-16; Supplementary Fig. 4].

(7) A validation on healthy donors should also have been made to show assay specificity.

We agree with the Reviewer. To validate the assay specificity, we processed blood samples collected from healthy donors and subjected them to the identical enrichment and staining processes applied on the clinical samples. The new results are presented in the revised manuscript [see p. 8, ¶ 2, lines 10-12].

(8) Regarding the RNA analysis of the cell clusters, there is no comparison with normal cells (leukocytes) and tumor cell line cells to proof that the RNA profiles they obtained are actual tumor cells. Better would have been whole genome sequencing.

We agree with the Reviewer on including positive and negative controls for our RNA sequencing studies. In the revised manuscript, we included clusters of two different prostate cancer cell lines (PC-3 and LNCaP) as well as leukocytes as positive and negative controls, respectively. Results from the sequencing of these controls are presented in the revised manuscript [see "Molecular analysis of patient CTC clusters" section: Fig. 5d; Supplementary Fig. 12, 13 & 14].

(9) I would also like to see a few more experiments to proof that the patient cell clusters are really viable (e.g., short term CTC culture).

We thank the Reviewer for this suggestion. Following the Reviewer's comment, we directly performed a viability assay on patient CTC clusters. In these new studies, we confirmed the patient CTC clusters enriched using our technology to include viable cells. These new results are presented in the revised manuscript [see p. 9, ¶ 3, lines 4-7; p. 19, ¶ 3, lines 1-11 and p. 20, ¶ 1, lines 1-4; Fig. 4h; Supplementary Fig. 11].

MINOR POINTS:

Figure 2f: The absolute number of leukocytes (i.e., not in percentage) should be also described in the text.

We added the absolute number of leukocytes to the revised manuscript [see p. 6, ¶ 2, lines 15-17].

In the material and methods section, the name of the antibody clones should be given.

We added the name of the antibody clones to the Methods section of the revised manuscript (see p. 18, ¶ 3, lines 5-11 & p. 19, ¶ 2, lines 3-4).

Lines 113-115 and reference to fig 2a: it is said in the manuscript text that cells were pre-labelled with fluorescent dyes and Fig2a shows a CK staining. These lines should be revised.

We thank the Reviewer for this comment. The text is edited accordingly in the revised manuscript to describe the fluorescent dyes in Fig. 2a (see p. 4, ¶ 3, lines 3-7).

Lines 200-202: the authors should precise how long the procedure takes.

When left unattended, we observed the microwells to dry in approximately 10 minutes. To allow more time for micromanipulation, we supplemented the setup with PBS to prevent drying when needed. We provided this information in the revised manuscript (see p. 19, ¶ 2, lines 9-10).

Reviewer #2:

Overall, I believe the authors have put together a comprehensive study on a unique microfilter device for CTC cluster isolation. While the scope of the paper is rather conservative, the authors address an important niche in the high throughput isolation of CTC clusters. The novelty of the microfilter design is profoundly unique from previous cluster isolation devices, however I do believe that other devices – particularly the continuous isolation microfluidic device developed by Ebb et al earlier this year – have achieved similar isolation efficiencies at comparable throughputs. That being said, the ease of operation of this microfilter combined with the downstream compatibility with single-cluster molecular analytics could potentially give this device a competitive edge and impact the field in a meaningful way.

The authors outline the fabrication and characterization of polymer microfilter able to isolate CTC clusters from multiple cancer types at high throughputs. The microfilter uses of a combination of silicon micromachining, soft lithography, and micro molding techniques, resulting in an efficient, cheap, and easily disposable plastic device. Notably, the filter is made of a photocuring, low-bind, fluorine-based polymer to reduce non-specific reactions. The microwell design is shown to have very high capture efficiency for CTC clusters comprised of greater than 2 CTCs per cluster. The wells are also shown to have a high cluster release efficiency and are easily accessible for individual micromanipulation of clusters. The authors validate their device functionality in both cell lines and patient samples. The authors also validate the downstream applicability of their microfilter by individually selecting 7 CTC clusters from two different prostate cancer patients to perform RNA sequencing. This analysis revealed both inter- and intra-patient heterogeneity in the gene expressions of the CTC clusters.

The authors do an excellent job of creating a device to specifically tackle the issue of CTC cluster isolation at a clinically relevant throughput. As established in the study, this microfilter provides a cheap, reproducible way to study CTC clusters and their biological relevancy in metastasis. The compatibility of the microfilter with micromanipulation and – by extension – single-cell RNA sequencing offers a very attractive mode to study cluster biology. However the authors should address the following minor concerns

We thank the Reviewer for their favorable comments and positive outlook for our manuscript.

1. The authors should discuss their device in the context of the other technologies, specifically the continuous flow ones

The novelty of this isolation device is based on the devices 1) throughput and 2) gentle flow conditions (i.e. the clusters experience shear stresses less than those found under physiological flow conditions). While the authors do an excellent job in showing the advantages of their device over the Cluster-Chip device (ref. 23), they only briefly discuss the benefits of their microfilter design over the continuous microfluidic device referenced in ref. 29. The authors claim the disadvantage of the of the continuous isolation device is that the higher flow rates threaten to damage cluster integrity (line 70-71). The authors of this study (Edds, J. F. et al), however, report a minimal loss in cluster integrity from their device. While the design of the two devices is very

different – one is a continuous, PDMS microfluidic device while the other is a microfabricated filter – they both claim a gentle recovery of CTC clusters under similar throughput.

As pointed out by the Reviewer, the microfluidic device developed by Edd, J. F. et al. was shown to process samples at high flow rates. However, it utilizes narrow (as low as 50 μm) microfluidic channels for cell sorting, which limits its ability to process and preserve larger clusters (e.g., >10 clusters). Large clusters are likely to clog the microfluidic channels or to dissociate under shear forces that are larger than physiological circulation pressures as they are forced to traverse thin microfluidic channels. As an example, the ovarian CTC cluster with >100 cells that we observed (see Fig. 4a, iii) would have been lost and/or damaged in such a microfluidic-based system due to its size (exceeding 0.5mm). To explicitly emphasize the advantage of our technology, we revised the manuscript accordingly [see p. 3, ¶ 1, lines 8-10]. In addition, our device has higher sensitivity to the mentioned system even at 100 mL/h which is significantly faster, also explicitly stated in our revised manuscript [see p. 6, ¶ 1, lines 1-3].

2. The authors do an excellent job in characterizing the capture and release efficiencies of their microfilter device. The authors also sufficiently address the viability of their released clusters from the devices following processing. One additional piece of evidence that would be helpful in supporting their claim is some kind of figure or statistic showing that the integrity of their clusters does not change between capture and release. The authors do an excellent job explaining the theory behind the advantages of flowing the wells in parallel, but they do not provide any data that I could find on how the distribution of cluster sizes changes from the start of processing, to capture and then release. This study would help to reinforce their claim that the filter operates under truly gentle conditions. Additionally, a simple simulation or mathematical explanation demonstrating the effective flow rate or shear stress through each well would be a helpful supplement to explanation.

Following the Reviewer's suggestions, we performed dissociation experiments to assess the integrity of the clusters during both capture and release. First, we performed dissociation test during capture under different flow rates (100, 250, 500, 750 mL/h). Briefly, we processed whole blood samples spiked with tumor cell clusters with the Cluster-Wells and simultaneously video recorded the spiked and missed (uncaptured) cell populations using a two-channel microfluidic interface. The captured (on-chip) cluster population was then obtained by scanning the device with a fluorescence microscope. Measured size (# of cells) distributions of unprocessed (spiked) and processed (missed + captured) populations were compared [see p. 5, ¶ 2, lines 7-9; Supplementary Fig. 5]. Measured distributions matched very well for 100 mL/h and 250 mL/h flow rates. We observed deviation between unprocessed and processed populations only at flow rates (>500 mL/h) much higher than the operational flow rate .

Similarly, we investigated the Cluster-Wells for potential dissociation of captured clusters during release. In these experiments, we first captured pre-stained clusters spiked in whole blood at the optimum flow rate (25 mL/h) and recorded the captured cell population using microfluidic interface imaged under a microscope. Following the capture, clusters were released into a Petri dish at 100X reverse flow rate (2500 mL/h) and counted with a fluorescence microscope. From comparisons between the measured size distributions of captured and released cluster

populations, we demonstrated that the integrity of cell clusters was preserved during release from the Cluster-Wells [see p. 7, ¶ 1, lines 5-7; Supplementary Fig. 7].

As suggested by the Reviewer, we also simulated fluid flow through Cluster-Wells using finite element analysis and calculated flow speed through a microwell [see Supplementary Fig. 1]. The simulation showed ~ 65 $\mu\text{m/s}$ maximum flow speed, which is ~10X lower than physiological free flow speed in human capillaries [1].

[1] Stücker, M. et al. Capillary blood cell velocity in human skin capillaries located perpendicularly to the skin surface: measured by a new laser Doppler anemometer. *Microvasc. Res.* 52, 188–192 (1996).

3. One additional piece of information that may be insightful would be to include patient treatment plan in the patient information table in the supplemental information. From the information provided, it is unclear at what point in the patient's treatment the blood is processed: are the patients treatment naïve, under treatment, or some combination of the two? Are any of the samples follow up samples? This information may help to provide further insight. For example, there seems to be a slight inverse correlation between CA125 levels and cluster abundance in ovarian patients; understanding the patients disease status and treatment path may help to explain potential trends.

We agree with the Reviewer that treatment information of the patients would be helpful for the readers. We revised the manuscript accordingly and provided all the available clinical data on the patient cohort [see Supplementary Table. 2 & 3].

Reviewer #3:

In the manuscript titled “High throughput, label-free isolation of circulating tumor cell clusters in meshed microwells”, Boya et al developed a microfabricated filtration chip for capturing rare CTC clusters. They perform a thorough validation of the system using several cell lines, and then show the ability to detect clusters from several patient samples, concluding with the RNA sequencing of 7 clusters from two patients with prostate cancer. The device design is a novel strategy for isolating CTC clusters, and they present data showing a 10-fold increase in effectiveness compared to a leading technology (Cluster-chip).

The strengths of this manuscript include using a novel fabrication process to make the mesh microwells for capturing CTC clusters and providing extensive characterization with both cell lines and detection of clusters in blood from patients with metastatic ovarian and prostate cancer. Although the authors provide sequencing of 7 individual clusters, there does not seem to be clear findings that emerge. Overall the manuscript validates the superiority of their technology to previous approaches but does not show how it can be exploited to reveal novel biological or clinical insights. Nevertheless, I recommend the manuscript is accepted provided the comments below are addressed.

We thank the Reviewer for favorable comments and positive outlook for our manuscript.

Major concerns:

1. The authors should include the published recovery rate and retrieval rate of Cluster-chip.

The data shown in our manuscript are based on the published recovery and retrieval rates of the Cluster-Chip. We explicitly stated this point in the revised manuscript [see p. 6, ¶ 1, line 1; p. 7, ¶ 2, lines 1-3 and p. 7, ¶ 2, line 7].

2. The authors should provide the distribution (and median size) of the clusters.

Based on the Reviewer’s comment, we have included the distribution profile and median size of clusters observed in prostate and ovarian cancer patients in the revised manuscript [see p. 9, ¶ 2, lines 6-7, Fig. 4f].

3. The claim that ascites clusters have larger cells has not been substantiated. The ascites staining is for EpCAM, a surface marker that stains the membrane and allows for clear size measurement. However, the CTC cluster has a staining for vimentin and cytokeratin, both of which are intercellular proteins and cannot be used by fluorescent imaging to measure size. A brightfield comparison or surface stain (e.g. EpCAM) would allow for a direct comparison of size of the cells. More data than the single image shown should be provided (e.g. measurements of the cells or multiple images). The number/concentration of clusters found in ascites should be included.

We agree with the Reviewer and revised our statement. In the revised manuscript, we compared the sizes of tumor cells isolated from blood and ascites sample from the patient using brightfield images and included these measurements [see p. 9, ¶ 1 lines 6-8; Fig. 4c; Supplementary Fig.

10. We also calculated the concentration of clusters found in the ascites sample and reported the results in the revised manuscript (see p. 9, ¶ 1, lines 3-6).

4. The presentation of the sequencing data is not entirely clear. For one patient, (Pca-2) none of the 5 clusters sequenced express any of the prostate markers (including the antibody used to identify the clusters- FOLH1/PSMA), and very few of the epithelial (and even mesenchymal markers). And the claim “Pca-2 CTC clusters presented similar levels of expression for epithelial and mesenchymal markers” is a little misleading, since the cells do not really express either class of genes. Non-specific binding of the antibody should be confirmed. What genes are the PCa-2 clusters high in?

We agree with the Reviewer. The testing of our antibodies used for identifying CTC clusters, as suggested by the Reviewer, confirmed their specificity for the target antigens (EpCAM and PSMA) on positive (LNCaP) and negative (T24) control cell lines.

Brightfield and fluorescence images of EpCAM- and PSMA-positive LNCaP prostate tumor cell clusters and EpCAM- and PSMA-negative T24 bladder tumor cell clusters isolated using Cluster-Wells. Both samples were subjected to the same staining protocol performed for patient samples except that the antibodies were applied separately. Scale bars, 20 μ m.

Considering a wider gene set could not robustly identify overexpression of tumor-associated genes. Given the CTC clusters isolated from Pca-1 and Pca-2 were sequenced in two separate runs using different facilities and the lack of sequencing controls in our original submission, we could not rule out potential artifacts in sequencing of CTC clusters from Pca-2 based on the available data. Therefore, we decided to remove the RNA-Seq data we obtained from PCa-2 clusters and sequenced CTC clusters from a new prostate cancer patient under the identical conditions used for sequencing PCa-1 clusters. Positive and negative controls were also included in the RNA-Seq process (1) to ensure against potential sequencing-associated artifacts and (2) to create a reference for identifying differentially expressed genes in CTC clusters.

5. How specific are the “metastasis” genes associated with cancer, as opposed to other biological processes? These are definitely genes related to cell cycle, but it is unclear where this list of genes came from (it is not found in either of the papers cited for the sentence [line 268- citations 9 and 13]). Also, why were only 4 genes selected to define “metastasis”?

These 4 genes were selected because they were reported to be upregulated with cancer metastasis [2-5] and were also expressed by clusters isolated from both patients in the original submission. However, we agree with the Reviewer that these genes are not specific to metastasis. In the revised manuscript, we present cluster expression of a wider set of genes (CHANDRAN_METASTASIS_TOP50_UP from the MSigDB database) that has been reported to be upregulated in metastatic prostate cancer and explicitly stated the source in the revised manuscript (see p. 11, ¶ 2, lines 20-22).

[2] Liang, W. et al. MARCKSL1 promotes the proliferation, migration and invasion of lung adenocarcinoma cells. *Oncol. Lett.* 19, 2272–2280 (2020).

[3] Banyard, J. et al. Identification of genes regulating migration and invasion using a new model of metastatic prostate cancer. *BMC Cancer* 14, 1–15 (2014).

[4] Maldonado, M. del M., Medina, J. I., Velazquez, L. & Dharmawardhane, S. Targeting Rac and Cdc42 GEFs in Metastatic Cancer. *Front. Cell Dev. Biol.* 8, 1–17 (2020).

[5] Struckhoff, A. P., Rana, M. K. & Worthylake, R. A. RhoA can lead the way in tumor cell invasion and metastasis. *Front. Biosci.* 16, 1915–1926 (2011).

6. In the discussion, there is again an overstatement of the “hybrid” epithelial/mesenchymal expression, when in fact there is quite low expression of both classes except for Vim and KRT10. In fact, one could argue that PCa-1 has high expression for both, while PCa-2 has low expression for both E/M.

We agree with the Reviewer and revised our statement (see p. 11, ¶ 2, lines 1-7). In our new cohort, we observed that all CTC clusters, including the new samples, expressed high levels of epithelial markers (EpCAM, CDH1, KRT8, KRT10, KRT18 and KRT19) while most of the mesenchymal markers were downregulated, except for Vimentin being expressed at low but detectable levels.

7. The statement “Furthermore, the lack of expression of AR, KLK3 (PSA), or PSMA in patient PCa-2 suggests that this patient could suffer from advanced, AR-negative prostate cancer” is entirely speculation, and is inconsistent with the fact that the sample stained positive for PSMA. A broader metastasis gene set would be helpful to confirm the nature of these clusters

We agree with the Reviewer and removed this statement in the revised manuscript. We also employed an expanded metastasis gene set from the literature to confirm the tumor origin of the studied patient clusters (see p. 11, ¶ 2, lines 20-22).

Additional comments that should be addressed:

1. Figure 2- c, d, and e do not have error bars. Were there any replicates?

We have performed replicate experiments (n=3) for these studies and revised the figures accordingly to demonstrate statistical variation.

2. To calculate retention rate of clusters, imaging was used of pre and post mesh blood. Is there a way to validate that clusters are not being broken up by the mesh? Or that any of the cells are sticking to the device (given the very low flow rate this could arise)? Also, how accurate was the image processing system (ie, if you run just in a loop without the chip, do you still detect the same numbers on both sides)?

We agree with the Reviewer that the accuracy of the image processing system should have been tested. To assess the accuracy of the system, we ran the two-channel microfluidic interface in a loop without the device attached. The cell numbers in both sides were tabulated (see Supplementary Table 1). The match between input and output microfluidic channels in the absence of device illustrated the reliability of the imaging setup used for characterization purposes. We also revised the text to report these results (see p. 5, ¶ 1, lines 6-8; p. 16, ¶ 1, lines 1-8).

To investigate whether the clusters were being broken up by the mesh, or that cells were lost due to sticking to the device, we directly imaged the retained population on the device along with the population at the inlet and outlet and compared size distributions of unprocessed cluster population that entered into the device (spiked) with the processed cluster population (captured + missed) under different flow rates (100, 250, 500, 750 mL/h) (see p. 5, ¶ 2, lines 7-9; Supplementary Fig. 5). We found that the sizes of unprocessed and processed cluster population matched for sample flow rates as high as 250 mL/h ruling out cluster damage at the nominal flow rate (25 mL/h) we operated our device.

3. Was there any evidence of large clusters breaking up into smaller clusters?

We observed dissociation of large clusters into smaller clusters only at flow rates significantly higher (>500 mL/h) than our operational flow rate (Supplementary Fig. 5). We revised the text accordingly to report the new data from our analysis on cluster dissociation (see p. 5, ¶ 2, lines 7-9).

4. All of the cell lines used appear to be adherent cell lines (ie, cell lines that do not naturally form in clusters). So “clusters” would just be different degrees of trypsinized cells? Would a cluster-forming suspension line produce different results? Also, what is the rough number of total detected clusters in the cell lines? Is this 1 cluster/mL (similar to patient data) or 10000 clusters/mL?

We subjected different cell lines used in this study to trypsin for a similar duration (~2 mins) followed by gentle pipetting. For all cell lines used in this study, pelleted cells were initially found to adhere to each other in groups of hundreds to thousands of cells following centrifugation and were subsequently dissociated mechanically into smaller clusters within the size range of interest for our study by pipetting. A revised explanation of this process is added to the Methods section of the revised manuscript (see p. 16, ¶ 2, lines 8-11).

A cell population adhering to each other more strongly would be more resilient to shear forces and would likely require higher flow rates to dissociate on our device but should not negatively impact the measured cluster capture rates.

In spiked cell experiments, the number of clusters spiked in simulated samples ranged 184 – 552 clusters. We provided this information in the revised manuscript (see p. 17, ¶ 1 lines 3-8).

5. A claim made that only at 750ul/min was dissociation seen (more single cells in waste than in input). This should be substantiated with data.

Based on the Reviewer's suggestion, we performed a systematic study on cluster dissociation as a function of sample flow rate. We reported the results of this new study to substantiate claims made on cluster dissociation at different flow rates in the revised manuscript (see p. 5, ¶ 2, lines 7-9; Supplementary Fig. 5).

6. The authors claim that there was a .05-.1% retention of WBCs in their samples. For a ~10mL sample, this would likely come out to ~50k-100k WBCs sticking in the chip (up to almost 1 per well). Where did the WBCs non-specifically bind (on the top region on in the wells)? Did WBCs come off during cluster retrieval? And would this make sequencing analysis difficult? And was there any platelet non-specific binding?

WBCs were observed to be adhering to the device at random locations. The majority of the WBCs were observed to remain adhered to the device under flow and they remained on the device during cluster retrieval. However, for RNA sequencing, we released CTC clusters micromanipulated from Cluster-Wells into an empty Petri dish and micromanipulated once more to ensure against WBC contamination. We explained these points in the revised manuscript (see p. 19, ¶ 2, lines 10-14).

As suggested by the Reviewer, we also analyzed platelet contamination on our devices and on average 0.023% of them were found to be non-specifically adhered to the device. We provided the new data in the revised manuscript (see p. 6, ¶ 2, lines 15-17).

7. For calculating retrieval rate, how did the total number (both retrieved and stuck clusters) compare to the expected value (as defined by your retention rate- from imaging the pre and post blood)?

We measured the cluster capture and retrieval (release) rates in different sets of experiments when characterizing our device. In the retrieval rate experiments, we only imaged the released and stuck populations. However, we do not expect the initial capture rate to be different for these retrieval experiments as we already characterized the device's capture performance under identical conditions (i.e., LNCaP cells at 25 mL/h).

8. Was viability of the true CTC clusters assessed? It is less surprising that a robust cell line would not be damaged by the processing.

Following the Reviewer's suggestion, we performed viability assay on patient-isolated CTC clusters and demonstrated there were viable cells within patient-derived CTC clusters. We presented new data in the revised manuscript (see p. 9, ¶ 3, lines 4-7; p. 19, ¶ 3, lines 1-11 and p. 20, ¶ 1, lines 1-4; Fig. 4h; Supplementary Fig. 11).

9. How long did it take for the wells to dry out during the micromanipulation?

When left unattended, the microwells were observed dry in ~10 minutes under tested conditions. To prevent this, we supplemented the container with PBS during micromanipulation. We revised the Methods section to provide this information and clarify the micromanipulation process employed in this study (see p. 19, ¶ 2, lines 9-10).

10. Line 234 claims 3/5 prostate cancer patients had clusters, but supplemental figure 1 appears to show that 5/7 patients had clusters.

We excluded clinical samples used for RNA-seq from enumeration studies as the RNA-seq samples were subjected to a different staining protocol exclusively targeting surface markers. For enumeration studies, we fixed the cells allowing us to target both surface and intracellular markers along with nuclei. We clarified this distinction in the revised manuscript (see p. 10, ¶ 2, lines 6-8; Supplementary Table 2 & 3).

11. A statement is made that “the developed CTC cluster assay requires only a commercial filter holder and no additional specialized hardware or training,” but it does require a syringe pump to operate as well (stated in the methods)

We agree with the Reviewer and revised the text accordingly (see p. 12, ¶ 1, line 14 and p. 13, ¶ 2, line 7).

Reviewers' Comments:

Reviewer #1:

Remarks to the Author:

The authors have addressed my comments and added new data which has improved the quality of the manuscript. I have no further comment.

Reviewer #2:

Remarks to the Author:

The authors sufficiently addressed the concerns raised by the reviewer regarding the original manuscript. The additional data support their conclusions and alleviated the concerns regarding the cluster disruption.

Reviewer #3:

Remarks to the Author:

the authors have addressed my comments and so I now recommend publication.